# Cnidarian hair cell development illuminates an ancient role for the class IV POU transcription factor in defining mechanoreceptor identity

**Ethan Ozment[1], Arianna N Tamvacakis[1], Jianhong Zhou[1], Pablo Yamild Rosiles-Loeza[2], Esteban Elías Escobar-Hernandez[2], Selene L Fernandez-Valverde[2], Nagayasu Nakanishi[1]\***

[1]Department of Biological Sciences, University of Arkansas, Fayetteville, United States; [2]Unidad de Genómica Avanzada (Langebio), Centro de Investigación y de Estudios Avanzados del IPN, Irapuato, Mexico

**Abstract** Although specialized mechanosensory cells are found across animal phylogeny, early evolutionary histories of mechanoreceptor development remain enigmatic. Cnidaria (e.g. sea anemones and jellyfishes) is the sister group to well-studied Bilateria (e.g. flies and vertebrates), and has two mechanosensory cell types – a lineage-specific sensory effector known as the cnidocyte, and a classical mechanosensory neuron referred to as the hair cell. While developmental genetics of cnidocytes is increasingly understood, genes essential for cnidarian hair cell development are unknown. Here, we show that the class IV POU homeodomain transcription factor (POU-IV) – an indispensable regulator of mechanosensory cell differentiation in Bilateria and cnidocyte differentiation in Cnidaria – controls hair cell development in the sea anemone cnidarian *Nematostella vectensis*. *N. vectensis* POU-IV is postmitotically expressed in tentacular hair cells, and is necessary for development of the apical mechanosensory apparatus, but not of neurites, in hair cells. Moreover, it binds to deeply conserved DNA recognition elements, and turns on a unique set of effector genes – including the transmembrane receptor-encoding gene *polycystin 1* – specifically in hair cells. Our results suggest that POU-IV directs differentiation of cnidarian hair cells and cnidocytes via distinct gene regulatory mechanisms, and support an evolutionarily ancient role for POU-IV in defining the mature state of mechanosensory neurons.

**\*For correspondence:**
nnakanis@uark.edu

**Competing interest:** The authors declare that no competing interests exist.

## Editor's evaluation

This study focusses on a little understood cell type, the hair cell , in the sea anemone *Nematostella vectensis*. It shows that the POU-IV transcription factor is required for the maturation of these likely mechanosensory neurons and activates a wide array of mechanosensory effector proteins. Because POU-IV transcription factors also play essential roles for the differentiation of mechanoreceptors in many bilaterian phyla, this suggests an evolutionarily ancient role of POU-IV in regulating mechanosensory identity.

## Introduction

One of the most fundamental sensory cell types that emerged in animal evolution is the mechanosensory cell – the specialized sensory epithelial cell that transduces mechanical stimuli (e.g. water vibration) into internal signals. These signals are then communicated, usually via the nervous system,

to effector cells (e.g. muscle cells) to elicit behavioral and/or physiological responses of the organism. Indeed, specialized mechanosensory cells are found across diverse animal lineages, from vertebrate hair cells, cephalopod angular acceleration receptors, to statocyst cells of cnidarian jellyfish and ctenophores. Typically, a mechanosensory cell bears an apical mechanosensory apparatus consisting of a single non-motile cilium surrounded by a circle of rigid microvilli with actin rootlets (i.e. stereovilli, or stereocilia), and extends basal neuronal processes that connect to the nervous system (reviewed in *Beisel et al., 2008*; *Budelmann, 1989*; *Manley and Ladher, 2008*).

The structure of animal mechanosensory cells is not uniform, however (reviewed in *Bezares-Calderón et al., 2020*). For instance, insect and cephalopod mechanosensory cells lack stereovilli (*Jarman, 2002*; *Budelmann, 1989*), while the apical mechanosensory apparatus of vertebrate hair cells is differently shaped, having a cilium on one side of a group of stereovilli of graded lengths, with the stereovilli next to the cilium being the longest (*Fain, 2003*). The observed morphological diversity in mechanosensory cells of distantly related animals has led to a fundamental question in animal mechanoreceptor evolution: whether the diversity evolved by divergence from a common ancestral form (*Beisel et al., 2008*; *Jørgensen, 1989*; *Schlosser, 2020*), or by independent evolution (*Coffin et al., 2004*; *Holland, 2005*). Addressing this question requires an understanding of the mechanisms of mechanoreceptor development across disparate groups of animals.

Developmental genetics of mechanosensory cells has been extensively studied in bilaterian models such as vertebrates and flies (reviewed in *Schlosser, 2020*; *Boekhoff-Falk, 2005*; *Beisel et al., 2008*). Yet, relatively little is known about the genetics of mechanoreceptor development in non-bilaterian, early-evolving animal groups such as Cnidaria (e.g. jellyfish, corals, and sea anemones), Ctenophora (combjellies), Placozoa and Porifera (sponges), the knowledge of which is key to defining the ancestral conditions for mechanoreceptor development basal to Bilateria. This baseline knowledge, in turn, is necessary for reconstructing how mechanoreceptors diversified in each lineage. In this paper, we focus on investigating the development of a fundamental, yet understudied, mechanosensory cell type of Cnidaria – the concentric hair cell.

Cnidaria is the sister group to Bilateria (*Medina et al., 2001*; *Putnam et al., 2007*; *Hejnol et al., 2009*; *Erwin et al., 2011*), and has two broad classes of mechanosensory cells – cnidocytes (*Brinkmann et al., 1996*) and concentric hair cells (*Arkett et al., 1988*; *Oliver and Thurm, 1996*; *Holtmann and Thurm, 2001*; not to be confused with vertebrate hair cells) – that are characterized by an apical mechanosensory apparatus consisting of a single cilium surrounded by a ring of stereovilli. The cnidocyte is the phylum-defining stinging cell type, and additionally contains a cnidarian-specific exocytotic organelle called the cnida (plural: cnidae) which is made up of a capsule enclosing a coiled tubule (reviewed in *Thomas and Edwards, 1991*; *Lesh-Laurie and Suchy, 1991*; *Fautin and Mariscal, 1991*). Cnidocytes are abundant in the ectodermal epithelium of cnidarian tentacles, and, upon perceiving mechanical stimuli, discharge cnidae by rapidly everting the coiled tubule to pierce nearby animals for defense and/or prey capture. There is no structural or functional evidence that the cnidocyte transmits sensory information to other cells, but firing of cnidae is thought to be modulated by neurons that innervate cnidocytes through chemical synapses (*Westfall, 2004*). Thus, the cnidocyte is a cnidarian-specific mechanosensory cell type that – uniquely among animal mechanosensory cells – functions as an effector cell.

The cnidarian hair cell, on the other hand, represents the classical mechanosensory cell type with dedicated sensory-neuronal function. Hair cells are integrated within the ectodermal epithelium of mechanosensory structures, such as gravity sensors of jellyfishes and tentacles of hydroids and corals (*Horridge, 1969*; *Lyons, 1973*; *Tardent and Schmid, 1972*; *Singla, 1975*; *Hündgen and Biela, 1982*). Structurally, the cnidarian hair cell exhibits the stereotypical mechanosensory neuron-like morphology described above, including the apical mechanosensory apparatus and basal neurites that become part of the basiepithelial nerve plexus (*Horridge, 1969*; *Singla, 1975*; *Singla, 1983*; *Hündgen and Biela, 1982*). Upon stimulation, the hair cells communicate mechanosensory information to other cells by converting mechanical stimuli into internal electrical signals (*Arkett et al., 1988*; *Oliver and Thurm, 1996*), and are thought to generate highly coordinated response behaviors such as righting and feeding. Similar to vertebrate hair cells, hair cells of jellyfish gravity sensors are sensitive to sound and can be lost due to noise trauma (*Solé et al., 2016*). Cnidarian hair cells show morphological and functional characteristics that parallel those of mechanosensory cells in other animal lineages, consistent with a deep evolutionary origin or convergent origins.

Although genetics of cnidocyte development is increasingly understood (e.g. *Babonis and Martindale, 2017*; *Richards and Rentzsch, 2015*; *Richards and Rentzsch, 2014*; *Wolenski et al., 2013*), that of cnidarian hair cell development remains poorly known. This knowledge gap severely limits our ability to reconstruct the evolutionary histories of mechanoreceptor development within Cnidaria and across the basal branches of the animal tree. A previous study has shown that the class IV POU homeodomain transcription factor (POU-IV or Brn-3)-encoding gene is expressed in the hair-cell-bearing mechanosensory organ called the touch plate in moon jellyfish *Aurelia sp.1* (*Nakanishi et al., 2010*), consistent with a role in cnidarian hair cell development. Yet, the function of POU-IV in cnidarian hair cell development, if any, remains undefined. As the first step toward elucidating the genetic mechanism of cnidarian hair cell development, here we dissect the role of POU-IV in the development of mechanosensory hair cells using the genetically tractable sea anemone cnidarian model *Nematostella vectensis*.

POU-IV is shared by all extant animal groups except for Ctenophora (comb jellies), indicative of early emergence in animal evolution (*Gold et al., 2014*). POU-IV is absent in choanoflagellates, although the class II POU-like gene has been reported to be present in *Mylnosiga fluctuans* indicative of a premetazoan origin of POU transcription factors (*López-Escardó et al., 2019*). As in other POU proteins, POU-IV is characterized by having a bipartite DNA-binding domain consisting of the N-terminal POU-specific domain and the C-terminal POU homeodomain (reviewed in *Herr and Cleary, 1995*). In Bilateria, POU-IV-binding DNA elements are POU-IV-class-specific and conserved; mammalian POU-IV proteins Brn3.0 (Brn-3a or POU4F1) and Brn3.2 (Brn-3b or POU4F2) and *Caenorhabditis elegans* POU-IV protein Unc-86 bind to a highly symmetrical core sequence AT(A/T)A(T/A)T(A/T)AT (*Gruber et al., 1997*). In bilaterian animal models such as *C. elegans,* POU-IV is known to function as a terminal selector – a transcription factor that determines mature cell identity via direct regulation of effector genes (reviewed in *Leyva-Díaz et al., 2020*). The cell type whose identity is defined by POU-IV across bilaterian lineages is the mechanosensory cell. In humans, mutations at one of the *pou-iv* loci – Brn-3c (Brn3.1 or POU4F3) – have been linked to autosomal dominant hearing loss (*Vahava et al., 1998*), and in Brn-3c knockout mice, auditory and vestibular hair cells fail to complete differentiation (*Erkman et al., 1996*; *Xiang et al., 1997b*) and are lost by cell death (*Xiang et al., 1998*). Likewise, in *C. elegans,* the *pou-iv* ortholog (*unc-86*) regulates differentiation of mechanosensory touch cells (*Chalfie and Sulston, 1981*; *Chalfie and Au, 1989*; *Finney and Ruvkun, 1990*; *Duggan et al., 1998*). In addition to its role in mechanoreceptor differentiation, POU-IV defines the identity of olfactory chemosensory neurons in *Drosophila* (*Clyne et al., 1999*), as well as retinal ganglion cells (Brn-3b; *Erkman et al., 1996*; *Gan et al., 1996*) and subsets of CNS neurons in mice (Brn-3a; *Serrano-Saiz et al., 2018*; *McEvilly et al., 1996*; *Xiang et al., 1996*). In Cnidaria, POU-IV is expressed not only in the developing mechanoreceptor of *Aurelia* sp.1 (*Nakanishi et al., 2010*) as described above, but also in the statocysts of the freshwater hydrozoan jellyfish *Craspedacusta sowerbii* (*Hroudova et al., 2012*). Also, POU-IV is required for postmitotic differentiation of cnidocytes, as well as elav::mOrange neurons, in *N. vectensis* (*Tournière et al., 2020*). Consistent with cnidarian POU-IV being a terminal selector, a genome-wide analysis of differential gene expression between POU-IV knockout mutant *N. vectensis* and their siblings indicates that POU-IV controls the expression of effector genes that define mature neural identity, such as those involved in ion channel activity (*Tournière et al., 2020*). However, it remains unknown if cnidarian POU-IV *directly* regulates effector gene expression, as expected for a terminal selector. Furthermore, although POU-IV recognition element-like sequences have been previously identified in the *N. vectensis* genome based on sequence similarity to bilaterian POU-IV-binding motifs (*Sebé-Pedrós et al., 2018*), cnidarian POU-IV recognition elements have not been experimentally defined, and consequently, whether the conservation of POU-IV-binding sequence extends beyond Bilateria remains unclear.

Sea anemones together with corals form the clade Anthozoa, which is sister to the Medusozoa – a group characterized by typically having a pelagic medusa (jellyfish) stage – consisting of Staurozoa, Hydrozoa, Scyphozoa, and Cubozoa (*Collins et al., 2006*; *Zapata et al., 2015*). Sea anemones have multicellular mechanosensory structures, known as the hair bundle mechanoreceptors, in the ectoderm of the oral feeding tentacles (*Mire-Thibodeaux and Watson, 1994*; *Mire and Watson, 1997*; *Watson et al., 1997*). A hair bundle mechanoreceptor consists of a central sensory cell surrounded by peripheral support cells (*Figure 1—figure supplement 1*). The central sensory cell exhibits morphological hallmarks of concentric hair cells, with an apical cilium surrounded by stereovilli, and basal

neurites. Support cells contribute stereovilli or microvilli that encircle the apical ciliary-stereovillar structure of the central hair cell. The cilium and stereovilli of the central cell and stereovilli/microvilli of support cells are interconnected by lateral linkages; in addition, extracellular linkages have been observed between the tips of stereovilli/microvilli of support cells, resembling the tip links of vertebrate mechanosensory hair cells (*Watson et al., 1997*). The apical sensory apparatus, or the hair bundle, of the mechanoreceptor thus consists of the cilium and stereovilli of the central hair cell and the peripheral stereovilli/microvilli of support cells (*Mire and Watson, 1997*). We note that in the literature, the support cells of hair bundle mechanoreceptors are sometimes referred to as hair cells (e.g. *Mire and Watson, 1997*). In this paper, in accordance with the morphological definition of cnidarian hair cells, a *hair cell* refers to the central sensory cell of the hair bundle mechanoreceptor, and a *support cell* refers to the cell that abuts the central sensory cell and contributes peripheral stereovilli/microvilli to the hair bundle.

In this report, we use the starlet sea anemone *N. vectensis* to investigate the role of POU-IV in the development of cnidarian hair cells. *N. vectensis* is a convenient model for studies of mechanisms of cnidarian development because of the availability of the genome sequence (*Putnam et al., 2007*) and a wide range of powerful molecular genetic tools including CRISPR-Cas9 genome editing (*Ikmi et al., 2014*; *Nakanishi and Martindale, 2018*). During embryogenesis, *N. vectensis* gastrulates by invagination to form an embryo consisting of ectoderm and endoderm separated by the extracellular matrix known as the mesoglea (*Kraus and Technau, 2006*; *Magie et al., 2007*). The blastopore becomes the mouth/anus ('oral') opening of the animal (*Fritzenwanker et al., 2007*; *Lee et al., 2007*). The embryo develops into a free-swimming, ciliated planula larva, which transforms into a polyp with circumoral tentacles that house mechanosensory hair cells in the ectoderm (*Figure 1—figure supplement 2*; *Nakanishi et al., 2012*; *Watson et al., 2009*). The polyp then grows and reaches sexual maturity. Previous studies have indicated that hair bundles of *N. vectensis* polyps are indeed sensitive to movement of surrounding water (*Watson et al., 2009*), and that stereovilli/microvilli of hair bundles express TRP (transient receptor potential)-like cation channels (*Mahoney et al., 2011*) and a putative extracellular linkage component cadherin 23 (*Watson et al., 2008*). In the present work, we provide evidence that POU-IV regulates postmitotic differentiation of hair cells by directly activating effector genes that define mature cell identity.

## Results

### Sea anemone hair cell has an apical cilium surrounded by a circle of stereovilli and extends basal neuronal processes

We first examined the structure of hair cells in the oral tentacles of the sea anemone *N. vectensis* at the primary polyp stage by light and electron microscopy. We used phalloidin to label F-actin enriched in stereovilli of hair cells, and the lipophilic dye DiI to label the plasma membrane of hair cells. Hair cells are an epithelial cell type whose cell body is pear-shaped and occurs exclusively in the superficial stratum of pseudostratified ectoderm in oral tentacles (*Figure 1*). The hair cell has an apical cilium surrounded by eight large-diameter stereovilli that extend actin filament-containing rootlets into the cytoplasm (*Figure 1B–I*). In primary polyps, the apical cilium is 10–15 µm long; stereovilli are 3–5 µm long and 200–400 nm in diameter; stereovillar rootlets are 2–3 µm long. Electron-lucent vesicles ranging from 50 to 100 nm in diameter are abundant in the cytoplasm of a hair cell (*Figure 1I*). Stereovilli of a hair cell are encircled by smaller-diameter microvilli (80–150 nm) contributed by adjacent support cells that are enriched in electron-dense vacuoles in the apical cytoplasm (*Figure 1E–I*). This multicellular apical sensory apparatus, consisting of the cilium and stereovilli of the hair cell surrounded by stereovilli/microvilli of support cells, constitutes the hair bundle (*Mire and Watson, 1997*). A subset of cnidocytes – nematocytes but not anthozoan-specific spirocytes – forms a morphologically similar apical mechanosensory apparatus known as the ciliary cone (*Fautin and Mariscal, 1991*); however, the ciliary cone of tentacular nematocytes in *N. vectensis* is less pronounced than that of hair cells, and consists of a single cilium surrounded by short microvilli (2–2.5 µm long) that lack actin rootlets (*Figure 1—figure supplement 3*). Basally, a hair cell extends thin neuronal processes that likely form synapses with the tentacular nerve net and/or longitudinal muscle fibers located at the base of the ectodermal epithelium alongside mesoglea (*Figure 1C, E and G*).

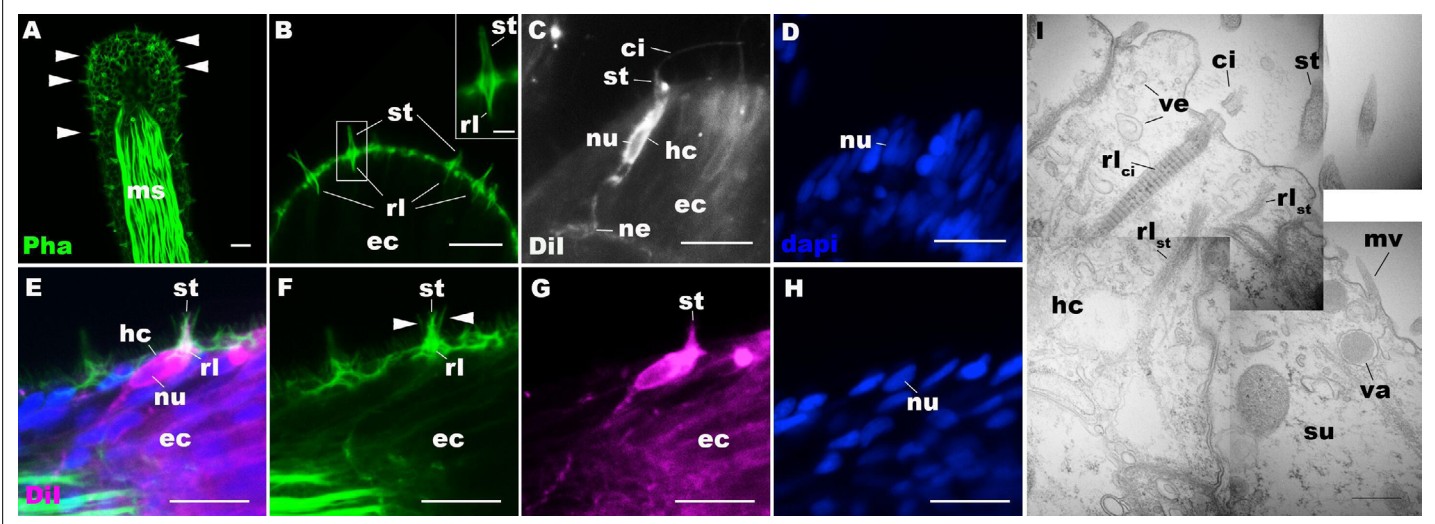

**Figure 1.** Morphology of sea anemone hair cells. (**A–F**) Confocal sections of oral tentacles of *Nematostella vectensis* at the primary polyp stage. Filamentous actin is labeled with phalloidin (Pha), and nuclei are labeled with DAPI (dapi). DiI is used to label cell membrane of a subset of hair cells. In A, the distal end of the tentacle is to the top, and in **B–I**, the apical surface of the ectodermal epithelium is to the top. **A**: sections through the tentacle. Numerous hair bundles (arrowheads) are evident on the tentacle surface. (**B**) Sections through the hair bundles at the tentacle tip, showing stereovilli (st) and their prominent rootlets (rl) of central hair cells. (**C–D**) Sections through a DiI-labeled hair cell (hc) at the tentacle tip. Note that the hair cell has an apical cilium (ci) surrounded at its base by stereovilli (st), and basally extended thin neurites (ne). An empty space within the cell body shows the location of a nucleus (nu), as evidenced by DAPI staining (D). (**E–H**) Sections through a DiI-labeled hair cell (hc) located near the tip of a tentacle. Arrowheads in F point to microvilli of the mechanoreceptor hair bundle contributed by peripheral support cells, which are DiI-negative. (**I**) Electron microscopic section of an apical region of the tentacular ectodermal epithelium of *N. vectensis* polyp, showing a hair cell (hc) and a support cell (su). The hair cell has stereovilli that extend dense filaments into the cytoplasm, forming 2–3 μm-long rootlets (rl$_{st}$), as well as numerous clear vesicles (ve), while the support cell has apical microvilli (mv) and electron-dense vacuoles (va). Abbreviations: ms muscle fibers; rl$_{ci}$ ciliary rootlet; ec ectoderm. Scale bar: 10 μm (**A–H**); 2 μm (inset in **B**); 500 nm (I).

The online version of this article includes the following figure supplement(s) for figure 1:

**Figure supplement 1.** Diagram of the hair bundle mechanoreceptor of sea anemones.

**Figure supplement 2.** Life cycle transition in the sea anemone cnidarian *Nematostella vectensis*.

**Figure supplement 3.** Morphology of nematocytes in the tentacles of *Nematostella vectensis* polyps.

**Figure supplement 4.** Hair cell development begins in the ectoderm of tentacle primordia at metamorphosis in sea anemones.

## Hair cells commence development at metamorphosis in the sea anemone

We next sought to determine the timing of hair cell development by using phalloidin to label stereovilli – the morphological hallmark of hair cells in *N. vectensis*. We never found stereovilli in the circumoral ectoderm during planula development (*Figure 1—figure supplement 4A,B*). However, pronounced stereovilli became evident in the circumoral ectoderm at the tentacle-bud stage (*Figure 1—figure supplement 4C,D*). These observations suggest that the hair cell is a postembryonic cell type that does not initiate development until metamorphosis in *N. vectensis*.

## Class IV POU transcription factor is postmitotically expressed in hair cells in the sea anemone

The *N. vectensis* genome contains a single gene that encodes the class IV POU homeodomain transcription factor (Nv160868; *Tournière et al., 2020*; *Nakanishi et al., 2010*; *Gold et al., 2014*), termed as *NvPOU4* by *Tournière et al., 2020*; in this paper, we will simplify the nomenclature by referring to *POU-IV/POU4/Brn3/unc-86* gene as *pou-iv* and its protein product as POU-IV. It has been previously shown that *pou-iv* mRNA is strongly expressed in circumoral ectoderm during metamorphosis in *N. vectensis* (*Tournière et al., 2020*), consistent with a role in tentacular morphogenesis. Although gene expression analysis using a transgenic reporter line has indicated that *pou-iv* is expressed in cnidocytes throughout the body including those in the tentacles (*Tournière et al., 2020*), whether *pou-iv*

is expressed in mechanosensory hair cells is not known. To address this, we first developed a rabbit polyclonal antibody against an N-terminal, non-DNA-binding region of the *N. vectensis* POU-IV based on the amino acid sequence predicted from *pou-iv* cDNA (see Materials and methods). As detailed in the next section, specificity of the antibody was confirmed by western blot analysis using *pou-iv* mutants and their wildtype siblings. In addition, immunostaining and in situ hybridization experiments showed that the pattern of anti-POU-IV immunoreactivity paralleled that of *pou-iv* mRNA expression (*Figure 2—figure supplement 1*), further supporting the specificity of the antibody. We therefore used immunostaining with the anti-POU-IV to analyze the expression pattern of POU-IV in developing oral tentacles of *N. vectensis* at metamorphosis. We found that POU-IV protein localized to nuclei of differentiating and differentiated hair cells, but not to those of support cells, in the ectoderm of developing tentacles (*Figure 2A–L*). In addition, we confirmed POU-IV expression in cnidocytes (*Figure 2—figure supplement 2*), consistent with the previous report (*Tournière et al., 2020*). Nuclear labeling by the anti-POU-IV was abolished when the antibody was preadsorbed with the POU-IV antigen prior to immunostaining (*Figure 2—figure supplement 3*), evidencing that the antibody reacts with nuclear POU-IV.

We then carried out EdU pulse labeling experiments to test whether any of the POU-IV-expressing cells in the tentacular ectoderm were at S-phase and thus proliferative. As observed for *pou-iv* transcript-expressing cells during embryogenesis (*Tournière et al., 2020*), we found that none of the POU-IV-expressing epithelial cells in the developing tentacles examined (n > 220 cells across three tentacle-bud-stage animals and eight primary polyps) incorporated EdU (e.g. *Figure 2M–P*), indicative of their postmitotic cell-cycle status. Taken together, the gene expression pattern suggests a role for POU-IV in postmitotic development of mechanosensory hair cells, as well as cnidocytes, in the tentacles of *N. vectensis* polyps.

## Generation of POU-IV mutant sea anemones

To investigate the function of POU-IV in hair cell development in *N. vectensis*, we generated a *pou-iv* mutant line by CRISPR-Cas9-mediated mutagenesis. First, a cocktail containing *pou-iv*-specific single guide RNAs (sgRNAs) and the endonuclease Cas9 protein was injected into fertilized eggs to produce founder (F0) animals. Multiple sgRNAs were designed to cleave flanking sites of the coding region of the *pou-iv* locus (*Figure 3A*; *Figure 3—figure supplement 1*). Large deletions were readily confirmed by genotyping PCR using genomic DNA extracted from single CRISPR-injected embryos (*Figure 3—figure supplement 1*). DNA sequencing of mutant bands confirmed that excision of both POU- and homeo-domains could be induced by this approach. F0 animals were raised and crossed with wildtype animals, in order to generate F1 heterozygous animals carrying a *pou-iv* knockout allele. Mutant allele carriers were identified by genotyping individual F1 polyps. One of the mutant alleles, which will be here referred to as *pou-iv⁻*, had a 705 bp deletion that removed a sequence encoding most of the POU domain (i.e. all but the first four N-terminal residues) and all of the homeodomain at the *pou-iv* locus (*Figure 3B*; *Figure 3—figure supplement 2*). This mutant allele differs from the previously generated *NvPOU4⁻* allele which harbors a frameshift mutation (31 bp deletion) at the start of the POU-domain-encoding sequence (*Tournière et al., 2020*). F1 *pou-iv* +/- heterozygotes were subsequently crossed with each other to produce F2 offspring, a quarter of which, on average, were *pou-iv* -/- mutants. *pou-iv* -/- mutants were identified by PCR-based genotyping methods (*Figure 3B and C*) using genomic DNA extracted from polyp tentacles (*Ikmi et al., 2014*) or from pieces of tissue isolated from early embryos (*Nakanishi and Martindale, 2018*; *Silva and Nakanishi, 2019*). Western blotting with the anti-POU-IV has confirmed that *pou-iv* -/- polyps express mutant POU-IV lacking DNA-binding domains (18.7 kDa), but not wildtype POU-IV (35.2 kDa) (*Figure 3D*), validating the specificity of the antibody.

## POU-IV is necessary for touch-response behavior of tentacles in the sea anemone

If POU-IV indeed plays a key role in postmitotic differentiation of mechanosensory hair cells, mechanosensitive behaviors of oral tentacles are expected to be perturbed in *pou-iv* null mutants. We tested this prediction by using F2 *pou-iv* -/- mutants and their heterozygous and wildtype siblings. In wildtype polyps, oral tentacles typically respond to touch stimuli by local contraction of longitudinal muscles. Strong and repeated touch stimuli of tentacles lead to excitation of longitudinal muscles in the body

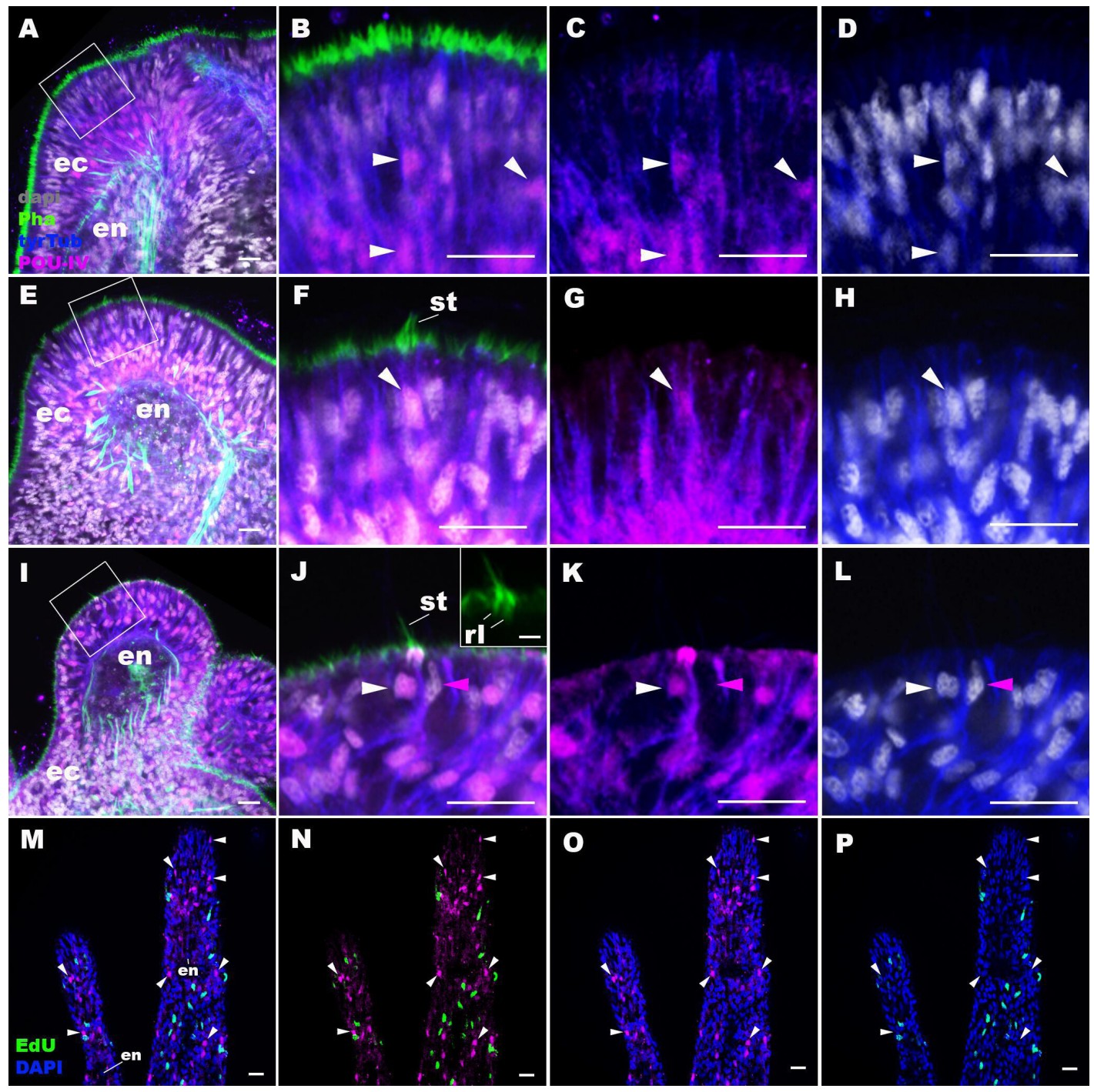

**Figure 2.** POU-IV is postmitotically expressed in hair cells of tentacular ectoderm at metamorphosis in the sea anemone. Confocal sections of *Nematostella vectensis* at metamorphosis, labeled with antibodies against POU-IV, and/or tyrosinated ∂-tubulin ('tyrTub'). Filamentous actin is labeled with phalloidin (Pha), and nuclei are labeled with DAPI (dapi). Proliferative cells are labeled by the thymidine analogue EdU. **A** shows a section through the presumptive tentacle primordia with the blastopore/mouth facing up. **E, I, M-P** show sections through developing oral tentacles with the distal end of the tentacle facing up; **M–P** are tangential sections of tentacles at the level of the surface ectoderm and parts of the endoderm (en). **B–D, F–H**, and **J–L** are magnified views of the boxed regions in **A, E**, and **I**, respectively, with the apical epithelial surface facing up. **A-D**: late planula. **E–H**: tentacle-bud. **I–P**: primary polyp. At the late planula stage prior to hair cell differentiation, POU-IV-positive nuclei are primarily localized at the basal and middle layers of the ectoderm of presumptive tentacle primordia (arrowheads in **B–D**); few POU-IV-positive nuclei are detectable at the superficial stratum. At the tentacle-bud stage, hair cells with pronounced stereovilli (st) and POU-IV-positive nuclei begin to develop in the superficial stratum of the ectodermal epithelium in tentacle primordia (arrowheads in **F–H**). POU-IV-positive nuclei in the superficial layer specifically occur in hair cells

*Figure 2 continued on next page*

*Figure 2 continued*

(white arrowheads in **J–L**) and not in adjacent support cells (purple arrowheads in **J–L**). The inset in **J** shows a magnified view of stereovilli (st) of a POU-IV-positive hair cell; note the presence of stereovillar rootlets (rl). In addition to hair cells, cnidocytes express POU-IV in the tentacular ectoderm (*Figure 2—figure supplement 2*; *Tournière et al., 2020*). POU-IV-positive cells are EdU-negative (arrowheads in **I–L**), evidencing their postmitotic cell-cycle status. Abbreviations: ec, ectoderm; en, endoderm. Scale bar: 10 μm (**A–P**); 2 μm (inset in **J**).

The online version of this article includes the following figure supplement(s) for figure 2:

**Figure supplement 1.** The pattern of immunoreactivity with an anti-POU-IV antibody recapitulates that of pou-iv mRNA expression.

**Figure supplement 2.** POU-IV localizes to the nuclei of cnidocytes in tentacular ectoderm of the sea anemone.

**Figure supplement 3.** Immunostaining with a preadsorbed anti-POU-IV antibody.

column, causing the tentacles to retract into the body column. In this study, a hair held in a tungsten needle holder was quickly depressed on the distal portion of each tentacle, and the presence/absence of the touch-response behavior of tentacles was scored for each animal; 100% of the F2 *pou-iv +/+* wildtype animals that were examined (n = 44) contracted at least one tentacle in response to touch (*Figure 4A and B*; *Figure 4—video 1*). In contrast, we observed that only 35% of the F2 *pou-iv -/-* knockout animals (n = 40) showed any sign of tentacular retraction in response to touch; 65% of the knockout mutants exhibited no discernable tentacular response to tactile stimuli (*Figure 4C and D*; *Figure 4—video 2*). The majority of F2 *pou-iv +/-* heterozygotes (87%, n = 62) showed touch-induced, tentacular responses; 13% did not show touch responses, suggestive of dose-dependent effects of POU-IV expression on the mechanosensory behavior. The reduced tentacular response to touch in *pou-iv -/-* mutants is not due to the inability of tentacular muscles to contract, as *pou-iv -/-* mutants responded to crushed brine shrimp extract by contracting tentacles (100%, n = 8 animals; *Figure 4—figure supplement 1*; *Figure 4—videos 3 and 4*). Hence, *pou-iv* is specifically required for the touch-sensitive behavior of oral tentacles in *N. vectensis*, consistent with POU-IV having a role in regulating the development of the mechanosensory hair cells.

## POU-IV is necessary for normal development of hair cells in the sea anemone

To understand the structural basis of touch insensitivity in *pou-iv* null mutants, we examined the morphology of tentacular cells in *pou-iv* null mutants and their heterozygous and wildtype siblings by light and confocal microscopy. At the primary polyp stage, F-actin labeling by phalloidin showed that the longitudinal muscles in the tentacle of F2 *pou-iv -/-* mutants developed normally (*Figure 5A, D and G*), consistent with the behavioral evidence demonstrating the ability of the mutant tentacles to contract in response to the brine shrimp extract. We have confirmed the previous finding that mature cnidocytes with cnidae fail to develop in *pou-iv* knockout mutants (*Figure 5C,F,I*; *Tournière et al., 2020*). In addition, we found that mature hair cells with stereovillar rootlets were lacking in the tentacles of F2 *pou-iv -/-* polyps (n = 6 animals), while mature hair cells formed normally in tentacles of *pou-iv +/-* (n = 6 animals) and *pou-iv +/+* (n = 3 animals) siblings (*Figure 5A–I*). Ciliary cone-like structures lacking stereovillar rootlets occurred in *pou-iv -/-* mutants (*Figure 5G–I*), raising the possibility that hair cells might undergo partial differentiation in *pou-iv -/-* mutants.

Electron microscopic observations confirmed these findings. Stereovillar rootlets and cnidae were absent in the tentacles of F2 *pou-iv -/-* polyps (n = 2 animals) but were present in the tentacles of their wildtype siblings (n = 2 animals) (*Figure 5J–L*; *Figure 5—figure supplements 1 and 2*). We also confirmed by electron microscopy the presence of a hair-cell-like cell that has an apical ciliary cone without stereovillar rootlets, surrounded by support cells with characteristic electron-dense vacuoles that contribute microvilli to the ciliary cone in *pou-iv -/-* mutants (*Figure 5K and L*); ciliary rootlets were observed in these hair-cell-like cells in *pou-iv -/-* mutants (*Figure 5—figure supplement 3*).

The lack of cnidae is consistent with the inability of *pou-iv* null mutants to capture prey as previously reported (*Tournière et al., 2020*), but cannot explain the lack of tentacular contraction in response to touch. Stereovillar rootlets provide stereovilli with structural resilience against physical damage and are necessary for normal mechanosensitivity in vertebrate hair cells (*Kitajiri et al., 2010*). We therefore suggest that touch insensitivity of oral tentacles in *pou-iv* null mutants results, at least in part, from the failure of hair cells to generate structurally robust apical mechanosensory apparatus (see Discussion).

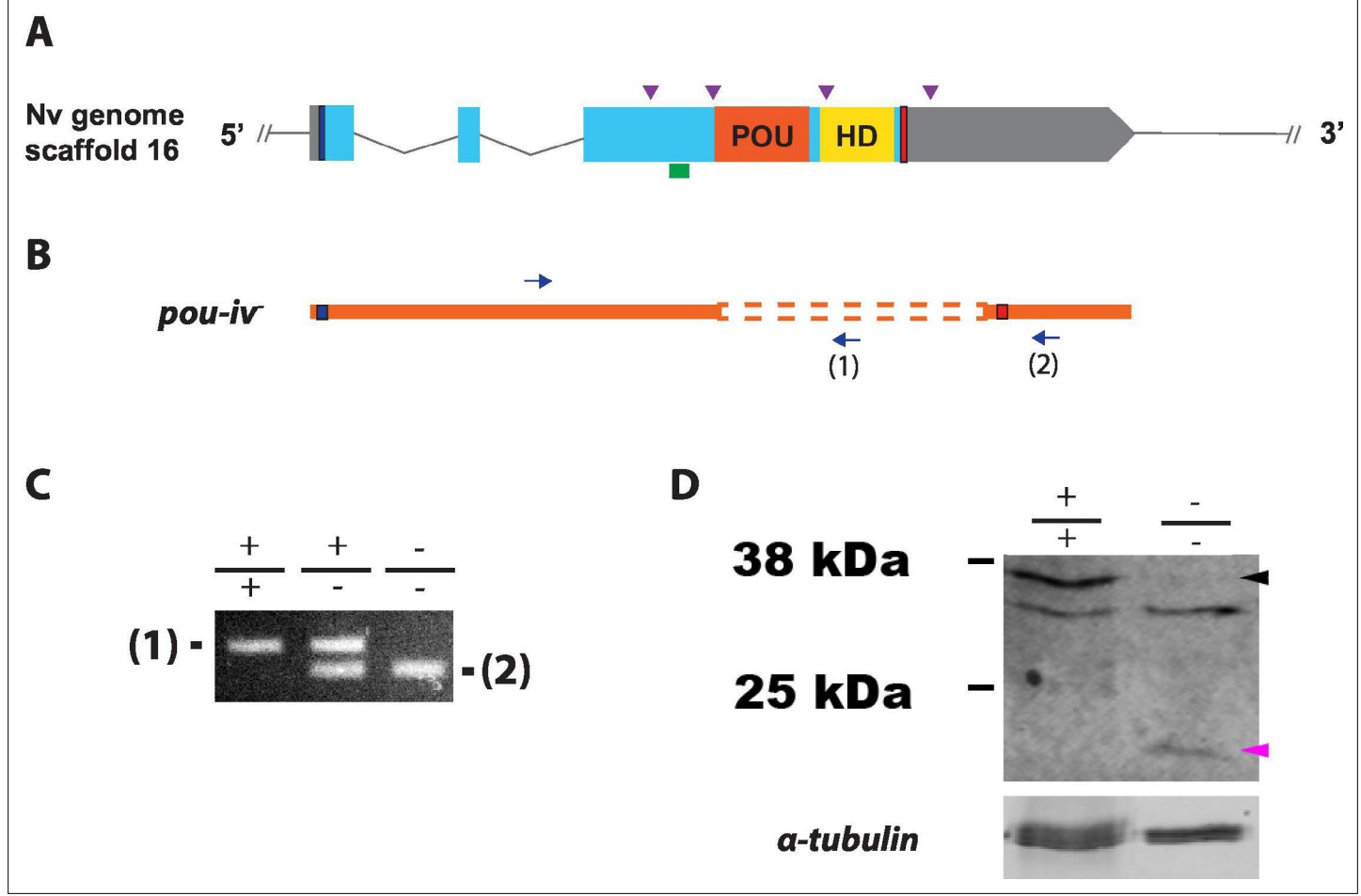

**Figure 3.** Generation of *pou-iv* null mutant sea anemones. (**A, B**) Diagrams of the *pou-iv* locus (**A**) and the disrupted mutant allele (*pou-iv⁻*; **B**). Blue bars show predicted translation start sites; red bars show predicted translation termination sites. In A, filled boxes indicate exons, and the regions that encode the POU- and homeo-domains are highlighted in orange ('POU') and yellow ('HD'), respectively. Purple arrowheads show single guide RNA (sgRNA) target sites. The region that encodes peptides targeted by the antibody generated in this study is indicated by a green line. In B, deletion mutation is boxed in dotted orange lines, and blue arrows mark regions targeted in the PCR analysis shown in C; reverse primers are numbered (1)–(2). (**C**) Genotyping PCR. Note that the wildtype allele-specific primer (1) generates a 689 bp PCR product from the wildtype allele ('+') but cannot bind to the *pou-iv⁻* allele due to deletion mutation. The primer (2) generates a 558 bp PCR product from the *pou-iv⁻* allele, and a 1312 bp PCR product from the wildtype allele. (**D**) Western blotting with an antibody against *Nematostella vectensis* POU-IV. An antibody against acetylated α-tubulin ('α-tubulin'; ca. 52 kDa) was used as a loading control. The anti-POU-IV reacts with a protein of expected size for wildtype POU-IV (35.2 kDa) in wildtype (+/+) polyp extracts, but not in *pou-iv* mutant (-/-) polyp extracts (black arrowhead). Also note that the antibody's reactivity with a protein of expected size for mutant POU-IV lacking DNA-binding domains (18.7 kDa) is detectable in mutant (-/-) extracts, but not in wildtype (+/+) extracts (purple arrowhead). The band just below the expected size of the wildtype POU-IV occur in both wildtype and mutant protein extracts, and therefore represents non-POU-IV protein(s) that are immunoreactive with the anti-POU-IV antibody.

The online version of this article includes the following source data and figure supplement(s) for figure 3:

**Source data 1.** An original gel image used to generate *Figure 3C* and the original image with relevant lanes labeled.

**Source data 2.** An original western blot image used to generate *Figure 3D* (top; anti-*Nematostella vectensis* POU-IV) and the original image with relevant lanes labeled.

**Source data 3.** An original western blot image used to generate *Figure 3D* (bottom; anti-acetylated α-tubulin) and the original image with relevant lanes labeled.

**Figure supplement 1.** Generation of pou-iv F0 mosaic mutants by CRISPR-Cas9-mediated mutagenesis in *Nematostella vectensis*.

**Figure supplement 1—source data 1.** An original gel image used to generate *Figure 3—figure supplement 1* (right) and the original image with relevant lanes labeled.

**Figure supplement 2.** Sequence alignment of wildtype and mutant *pou-iv* alleles.

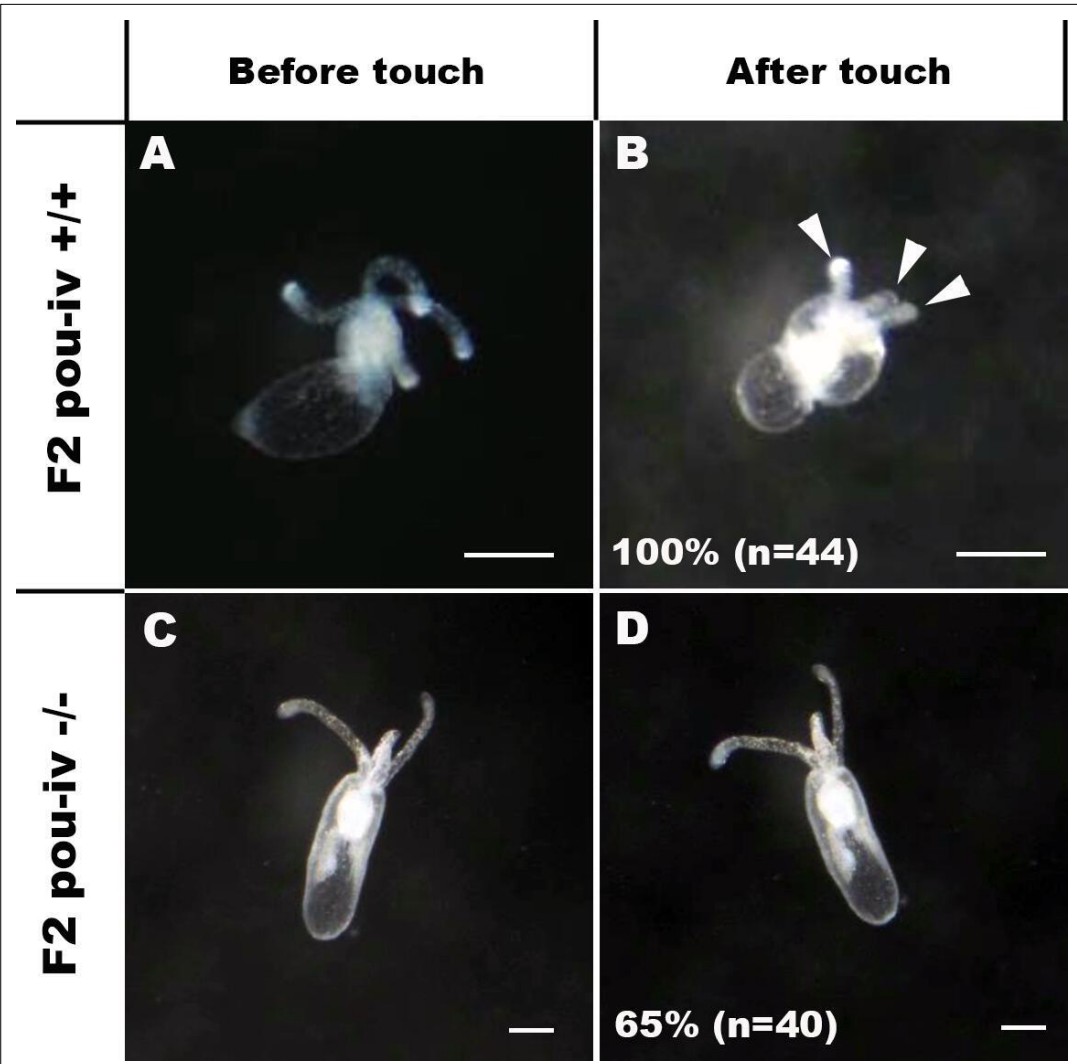

**Figure 4.** POU-IV is essential for touch-response behavior in the sea anemone. (**A-D**) Behavior of wildtype (F2 *pou-iv* +/+, **A, B**) and mutant (F2 *pou-iv* -/-, **C, D**) *Nematostella vectensis* polyps in response to tactile stimuli to their oral tentacles. A hair held in a tungsten needle holder was used to touch the distal portion of each tentacle. Animals before (**A, C**) and after (**B, D**) tentacle touch are shown. Tactile stimuli to tentacles elicit tentacular retraction in the wildtype individual (100%, n = 44; **A, B**). In contrast, the majority of *pou-iv* homozygous mutants were touch-insensitive (65%, n = 40; **B, D**); only 35% of the animals showed any contractile response to touch stimuli. Arrowheads in **B** point to retracted tentacles. Scale bar: 1 mm.

The online version of this article includes the following video and figure supplement(s) for figure 4:

**Figure supplement 1.** *pou-iv* mutants respond to brine shrimp extract by tentacular contraction and pharyngeal protrusion.

**Figure 4—video 1.** Touch-sensitive behavior of a wildtype (F2 *pou-iv* +/+) polyp.
https://elifesciences.org/articles/74336/figures#fig4video1

**Figure 4—video 2.** Touch-insensitive behavior of a *pou-iv* mutant (F2 *pou-iv* -/-) polyp.
https://elifesciences.org/articles/74336/figures#fig4video2

**Figure 4—video 3.** Behavior of a wildtype (F2 *pou-iv* +/+) polyp upon exposure to brine shrimp extract.
https://elifesciences.org/articles/74336/figures#fig4video3

**Figure 4—video 4.** Behavior of a *pou-iv* mutant (F2 *pou-iv* -/-) polyp upon exposure to brine shrimp extract.
https://elifesciences.org/articles/74336/figures#fig4video4

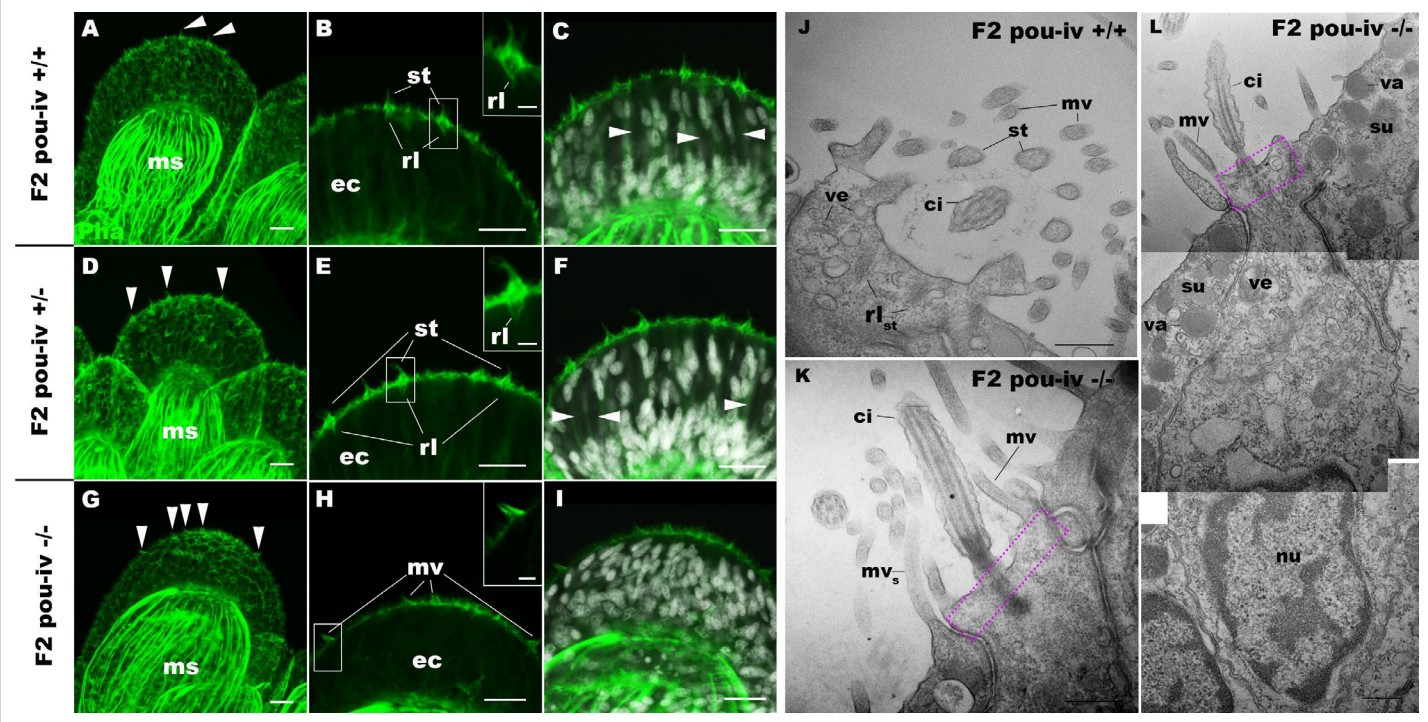

**Figure 5.** POU-IV is necessary for hair cell differentiation in the sea anemone. (**A-I**) Confocal sections of oral tentacles of wildtype (F2 *pou-iv +/+*, **A-C**), heterozygous (F2 *pou-iv +/-*, **D-F**), and homozygous *pou-iv* mutant (F2 *pou-iv -/-*, **G-I**) *Nematostella vectensis* polyps. Filamentous actin is labeled with phalloidin (Pha), and nuclei are labeled with DAPI (dapi). In all panels, the distal end of the tentacle is to the top. (**A, D, G**) Sections through the tentacle. (**B, C, E, F, H, I**) Sections through hair bundles/ciliary cones at the tip of tentacles. Ciliary cones occur on the epithelial surface of the tentacle regardless of the genotype (arrowheads in **A, D, G**). However, stereovilli (st) with rootlets (rl) characteristic of mechanosensory hair cells are observed in wildtype (**B**) and heterozygous (**E**) siblings, but not in homozygous *pou-iv* mutants whose ciliary cones contain microvilli without prominent actin rootlets (mv in **H**). Arrowheads in C and F indicate spaces occupied by cnidocysts in wildtype and heterozygous siblings, respectively, which are absent in *pou-iv* homozygous mutants (**I**; *Figure 5—figure supplement 1*). (**J–L**) Electron microscopic sections of a hair cell of a F2 *pou-iv +/+* polyp (**J**) and an epithelial cell with hair-cell-like morphologies in an F2 *pou-iv -/-* polyp (**K, L**). In all panels, apical cell surfaces face up. K and L are sections of the same cell at different levels. The hair-cell-like epithelial cell of the mutant has a central apical cilium surrounded by a collar of rootlet-less microvilli (mv in **K, L**), which are encircled by microvilli of the adjacent support cells (mv$_s$ in L), forming a ciliary cone. It also has numerous clear vesicles (ve in **L**) in the cytoplasm, characteristic of hair cells (ve in **J**; *Figure 1G*). Support cells of mutants are morphologically indistinguishable from those of wildtype animals, having characteristic large electron-dense vacuoles (va in **L**) in addition to apical microvilli (mv$_s$ in L) that contribute to the ciliary cone/hair bundle. Consistent with light microscopy data (**A–C, G–I**), stereovillar rootlets (rl$_{st}$) are absent in the F2 *pou-iv -/-* polyp, but are present in hair cells of their wildtype siblings (**J**). In K and L, regions of apical cytoplasm where stereovillar rootlets would normally be observed are boxed with dotted purple lines. Abbreviations: ms muscle fibers; ec ectoderm; st stereovilli; ci cilium; rl$_{st}$ stereovillar rootlets. Scale bar: 10 µm (**A-I**); 2 µm (insets in **B, E, H**); 500 nm (**J–L**).

The online version of this article includes the following figure supplement(s) for figure 5:

**Figure supplement 1.** *pou-iv* mutants lack mature cnidocytes.

**Figure supplement 2.** F2 *pou-iv* wildtype siblings develop hair cells with stereovillar rootlets.

**Figure supplement 3.** Hair-cell-like cells of *pou-iv* mutants have ciliary rootlets.

## POU-IV is necessary for maturation, but not initial differentiation or survival, of hair cells in the sea anemone

The lack of functional hair cells in *pou-iv -/-* mutants is consistent with POU-IV having a necessary role in initial differentiation and/or maturation of hair cells. In order to more precisely define the functional role of POU-IV in hair cell development, we investigated the morphological and molecular characteristics of epithelial cells expressing the mutant form of POU-IV, which we refer to as POU-IV(-), in tentacular ectoderm of *pou-iv -/-* mutants. Because the epitope that the anti-POU-IV antibody reacts with is intact in the protein encoded by the *pou-iv⁻* allele (*Figure 3A, B and D*), it was possible to use immunostaining with the anti-POU-IV to localize POU-IV(-) in *pou-iv -/-* mutants. A number of epithelial cells in the tentacular ectoderm were found to express POU-IV(-) (*Figure 6A–C*). In contrast to the primarily nuclear localization of POU-IV in wildtype animals (*Figure 2*), however,

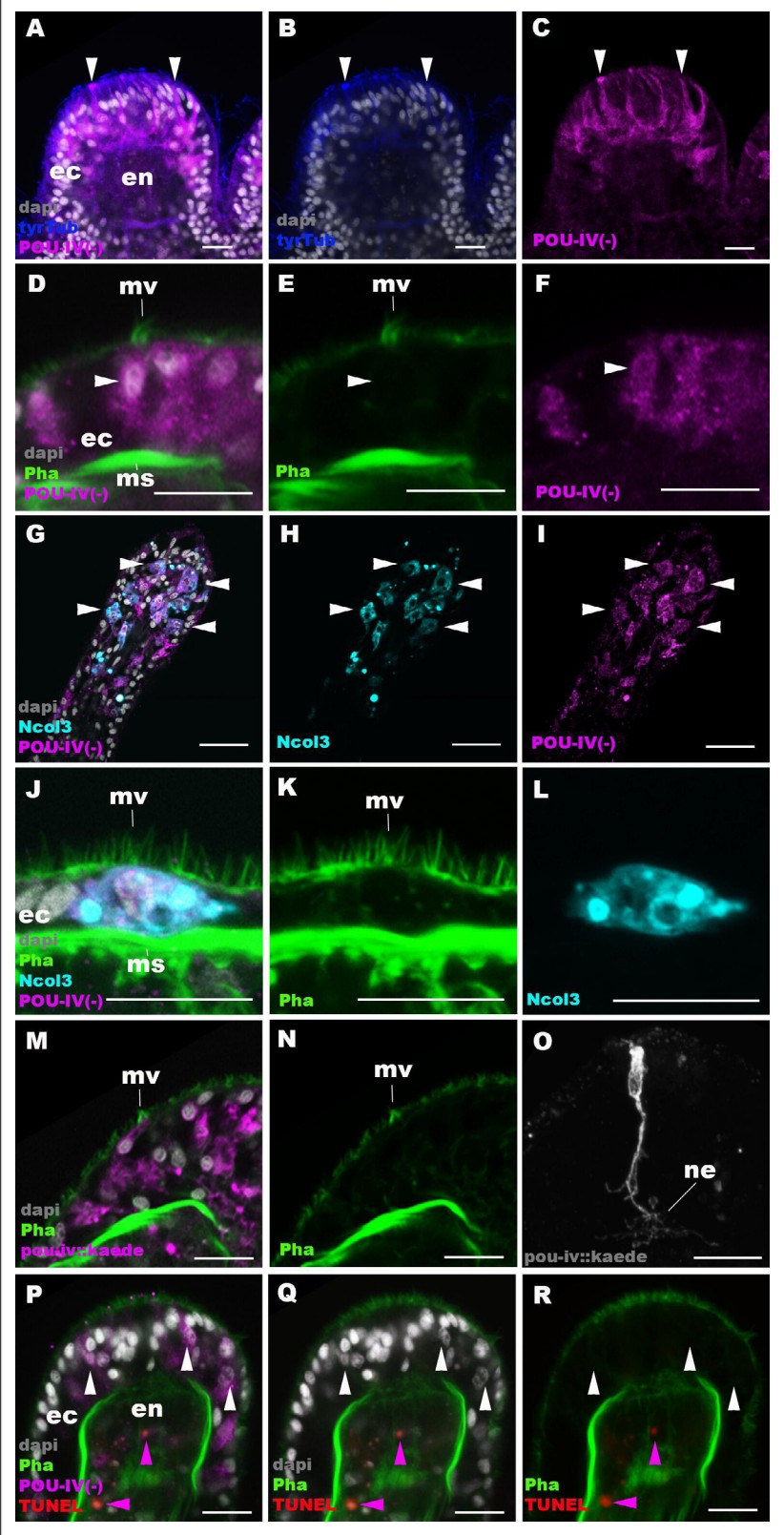

**Figure 6.** POU-IV is necessary for maturation of hair cells in the sea anemone. Confocal sections of oral tentacles in F2 *pou-iv -/- Nematostella vectensis* polyps, labeled with antibodies against tyrosinated ∂-tubulin ('tyrTub'), minicollagen 3 ('Ncol3'; *Zenkert et al., 2011*), mutant POU-IV ('POU-IV(-)'), and/or Kaede fluorescent protein ('pou-iv::kaede'). DNA fragmentation is labeled by terminal deoxynucleotidyl transferase dUTP nick end-labeling

*Figure 6 continued on next page*

*Figure 6 continued*

(TUNEL). Filamentous actin is labeled with phalloidin (Pha), and nuclei are labeled with DAPI (dapi). In all panels, the apical surface of the tentacular ectodermal epithelium is to the top. (**A–C**) Sections through developing oral tentacles with the distal end of the tentacle facing up. Arrowheads point to a subset of POU-IV(-)-expressing epithelial cells, which are abundant in the tentacular ectoderm (ec). Note the cytoplasmic distribution of the POU-IV(-) likely resulting from the lack of nuclear localization signal. (**D–F**) Sections showing ciliary cone microvilli (mv)-bearing cells. Ciliary cone-bearing epithelial cells express POU-IV(-) (arrowheads). (**G–I**) Sections at the level of surface ectoderm of developing oral tentacles with the distal end of the tentacle facing up. A subset of POU-IV(-)-expressing cells are Ncol3-positive (arrowheads), representing immature cnidocytes. (**J–L**) Sections showing an immature cnidocyte which expresses POU-IV(-) and Ncol-3. Note that the cell bears apical microvilli (mv) that do not form a ciliary cone. (**M–O**) Sections showing immature hair cells in F2 *pou-iv -/- N. vectensis* polyps injected with *pou-iv::kaede* construct. Note the presence of ciliary cone microvilli (mv) and basal neurites (ne). (**P–R**) Sections through tentacles with the distal end facing up. White arrowheads point to nuclei of POU-IV(-)-expressing ectodermal epithelial cells, which are TUNEL-negative. TUNEL-positive, pyknotic nuclei are observed in the endoderm (purple arrowheads). Abbreviations: ec ectoderm; en endoderm; ms muscle fiber. Scale bar: 10 μm.

The online version of this article includes the following figure supplement(s) for figure 6:

**Figure supplement 1.** *pou-iv::kaede* reporter construct drives transgene expression in hair cells and cnidocytes.

**Figure supplement 2.** Cnidocytes in *pou-iv* null mutants develop basal processes.

POU-IV(-) is distributed throughout the cytoplasm of POU-IV(-)-expressing cells in *pou-iv -/-* mutants (*Figure 6A–F*), presumably due to the lack of nuclear localization signal (located at the N-terminal end of the homeodomain; *Sock et al., 1996*) in POU-IV(-) (*Figure 3B*). We found that the epithelial cells bearing apical ciliary cones in *pou-iv -/-* mutants expressed POU-IV(-) (*Figure 6D–F*) and therefore could represent partially differentiated hair cells that failed to undergo maturation. Alternatively, as ciliary cones characterize nematocytes in wildtype *N. vectensis*, it was possible that these ciliary cone-bearing epithelial cells in *pou-iv -/-* mutants were immature nematocytes without cnidae.

To clarify the identity of ciliary cone-bearing epithelial cells in *pou-iv -/-* mutants, we used an anti-body against a pan-cnidocyte differentiation marker minicollagen 3 (Ncol3; *Babonis and Martindale, 2017*; *Zenkert et al., 2011*) to label immature cnidocytes. It was previously shown that Ncol3 was expressed in a subset of ectodermal epithelial cells of *pou-iv* knockout mutants despite the lack of mature cnidae, indicating that immature cnidocytes are present in *pou-iv* mutants and that *pou-iv* is not necessary for initial differentiation of cnidocytes (*Tournière et al., 2020*). By using immunostaining with an anti-Ncol3, we confirmed that Ncol3-positive immature cnidocytes in *pou-iv -/-* mutants indeed expressed POU-IV(-) (*Figure 6G–I*). However, none of the Ncol3-positive immature cnidocytes in *pou-iv -/-* mutants had distinct apical ciliary cones (e.g. *Figure 6J–L*), suggesting that ciliary cone-bearing epithelial cells in *pou-iv -/-* mutants represent immature hair cells, and not immature nematocytes. Thus, hair cells appear to be present in their immature, yet morphologically differentiated, form in *pou-iv -/-* mutants. The presence of partially differentiated hair cells in *pou-iv -/-* mutants supports the hypothesis that POU-IV regulates maturation, but not initial differentiation, of hair cells in *N. vectensis*.

As discussed above, the absence of stereovillar rootlets in hair cells of *pou-iv -/-* mutants may underlie the observed touch insensitivity of the mutants. It was also possible that these immature hair cells failed to extend basal neurites to form normal mechanosensory neural circuits. To examine this possibility, we visualized the morphology of immature hair cells in *pou-iv -/-* mutants by using a *pou-iv::kaede* transgenic reporter construct, in which the 3.2 kb genomic sequence upstream of the start codon of the *pou-iv* gene was placed in front of the Kaede fluorescent protein-encoding gene (*Ando et al., 2002*). We first confirmed that the *pou-iv::kaede* reporter construct indeed drove the expression of Kaede in POU-IV-positive cell types – hair cells and cnidocytes – in tentacular ectoderm of wildtype animals, recapitulating the endogenous POU-IV expression pattern (*Figure 6—figure supplement 1*). Interestingly, we unexpectedly found that cnidocytes, in addition to hair cells, had basal neurite-like processes (*Figure 6—figure supplement 1I-L*), which has never been reported in cnidarian litera-ture to our knowledge. We then injected *pou-iv::kaede* plasmids into *pou-iv* F2 zygotes, which were allowed to develop into primary polyps, and subsequently carried out immunostaining with antibodies against Kaede and Ncol3. Animals lacking mature cnidae based on Ncol3 staining were assumed to be *pou-iv -/-* mutants. In these presumptive mutants, Kaede-positive immature hair cells were readily

identifiable based on morphology and position; their cell bodies were pear-shaped and located in the superficial stratum of the tentacular ectoderm, some of which contained apical microvilli that are organized into a ciliary cone-like microvillar structure (*Figure 6M and N*). These immature hair cells, however, developed morphologically normal basal neurites (*Figure 6O*), indicating that *pou-iv* is not necessary for neurite extension in hair cells. Neither is *pou-iv* required for the development of basal neurite-like processes in cnidocytes; basal processes were observed in Ncol3-positive immature cnidocytes (*Figure 6—figure supplement 2*).

In mice, one of the *pou-iv* paralogs – *brn3c* – is thought to be required for survival of hair cells because the number of apoptotic cells increases in the inner ear sensory epithelia in Brn-3c null mutant mice (*Xiang et al., 1998*). We have therefore tested whether *pou-iv* regulates hair cell survival in *N. vectensis*, by carrying out the terminal deoxynucleotidyl transferase dUTP nick end-labeling (TUNEL) assay in *pou-iv -/-* mutants. We found that none of the POU-IV(-)-expressing epithelial cells examined in the tentacular ectoderm (n = 100 cells across five primary polyps) had TUNEL-positive, pyknotic nuclei indicative of apoptotic DNA fragmentation, although TUNEL-positive nuclear fragments were commonly observed in the endoderm (*Figure 6P–R*). Thus, in sea anemones, POU-IV does not appear to be directly involved in the survival of hair cells.

## POU-IV-binding motifs are conserved across Cnidaria and Bilateria

The evidence presented above thus indicates that in *N. vectensis,* POU-IV is involved in the maturation of mechanosensory hair cells – in addition to that of cnidocytes (*Tournière et al., 2020*). How, then, does POU-IV control the development of these two distinct mechanosensory cell types? One possibility is that the POU-IV transcription factor regulates the expression of a shared set of genes critical for differentiation of both cell types. Given that both hair cells and cnidocytes are mechanosensory, POU-IV might control the expression of the same set of mechanotransduction genes in these cell types. Another possibility is that POU-IV regulates the expression of distinct sets of genes in different neural cell types, actively driving the differentiation of the two mechanosensory cell types.

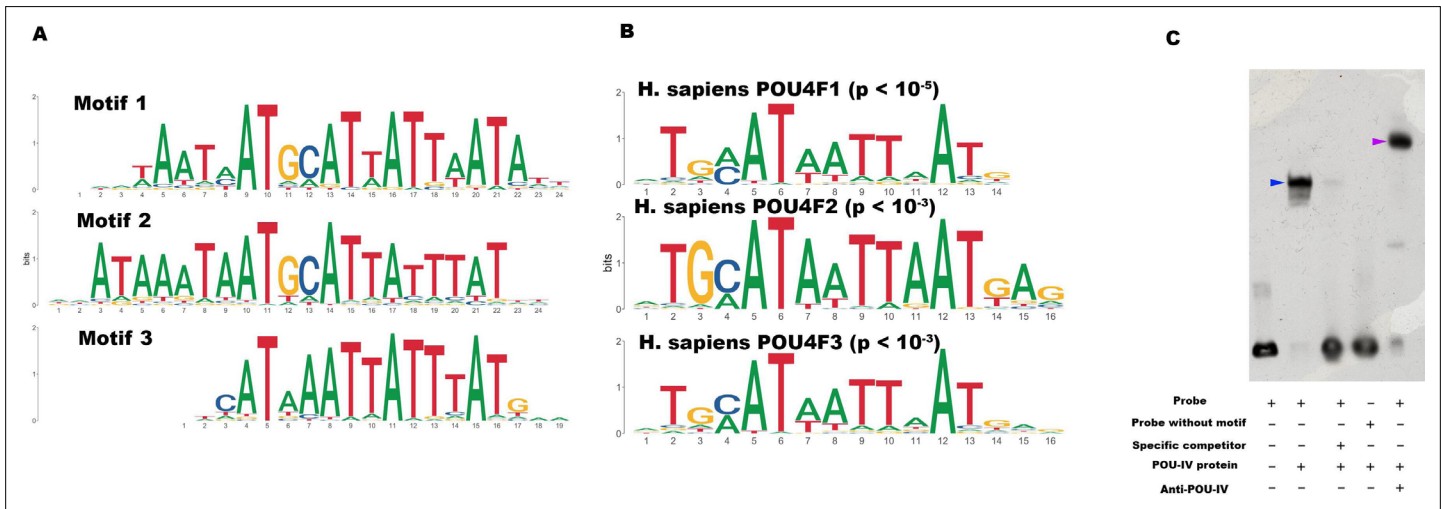

**Figure 7.** POU-IV-binding motifs are conserved across Cnidaria and Bilateria. (**A**) Motifs enriched in *Nematostella vectensis* POU-IV chromatin immunoprecipitation sequencing (ChIP-seq) peaks. (**B**) *Homo sapiens* POU motifs resulting from sequence alignment and comparison against the Jaspar database. The p-value reported corresponds to the highest p-value for any of the three *N. vectensis* POU4 motifs found. (**C**) Electrophoretic mobility shift assay (EMSA) using purified *N. vectensis* POU-IV protein and a 50 bp DNA probe containing the conserved core motif *CATTATTAAT*. Note that retardation of probe migration occurs in the presence of POU-IV protein (blue arrowhead; lane 2), indicative of formation of the protein-DNA complex. Retardation is inhibited in the presence of an unlabeled competitor probe ('specific competitor'; lane 3). Removal of the motif sequence in the probe ('probe without motif') abolishes retardation of probe migration by POU-IV (lane 4), demonstrating that the motif is necessary for formation of the protein-DNA complex. The mobility of the probe is further decreased in the presence of the anti-POU-IV antibody (purple arrowhead; lane 5), confirming that the protein bound to the probe is POU-IV.

The online version of this article includes the following source data for figure 7:

**Source data 1.** List of 12,972 genome-wide-binding sites for POU-IV.

**Source data 2.** An original gel image used to generate *Figure 7C* and the original image with relevant lanes labeled.

To begin to address this question, we identified genome-wide-binding sites for POU-IV by chromatin immunoprecipitation sequencing (ChIP-seq) using the antibody against *N. vectensis* POU-IV. We used adult *N. vectensis* for this experiment, because (1) neurogenesis continues through adulthood (e.g. *Havrilak et al., 2021*), and (2) we needed over 1 g of tissue samples, which was more difficult to obtain from other developmental stages including primary polyps. We sequenced anti-POU-IV immunoprecipitated DNA and input DNA, and mapped the reads to the *N. vectensis* genome (*Putnam et al., 2007*). We identified 12,972 genomic sites that were enriched in ChIP DNA (i.e. ChIP peaks) (*Figure 7—source data 1*). We then performed a de novo motif search and motif enrichment analysis, and found three motifs *rwrwaatmatgc**attattaat**att* (motif 1; E = 5.2e-075), *rmataaataatgc**attatttat**ky* (motif 2; E = 1.2e-052), and *tkcataa**ataatttat**gmm* (motif 3; E = 4.8e-36) that were enriched toward the center of ChIP peaks (p = 3.5e-368 and p = 1.0e-138, respectively) (*Figure 7A*). When we compared these three motifs against the Jaspar database (*Fornes et al., 2020*), we discovered that they showed significant sequence similarity to *Homo sapiens* POU4F1-, POU4F2-, and POU4F3-binding motifs (*Figure 7B*; $p < 10^{-5}$, $p < 10^{-3}$, and $p < 10^{-3}$, respectively), indicative of deep conservation of POU-IV-binding motifs across Cnidaria and Bilateria. Indeed, the motifs we have identified contain the sequence AT(A/T)ATT(A/T)AT (shown in bold in motif sequences above), which is nearly identical to the core recognition sequence of bilaterian POU-IV, AT(A/T)A(T/A)T(A/T)AT (*Gruber et al., 1997*). In addition, the preference of GC residues 5′ to the core recognition sequence is evident in motifs 1 and 2 (underlined in motif sequences above), and in bilaterian POU-IV-binding sequences (*Gruber et al., 1997*), and therefore appears to be conserved. We tested the ability of POU-IV to bind to the core recognition motif-like sequences by electrophoretic mobility shift assays (EMSAs), and confirmed that they were indeed essential for DNA recognition by POU-IV (*Figure 7C*). We infer that in the last common ancestor of Cnidaria and Bilateria, POU-IV bound to the consensus DNA element GCAT(A/T)ATT(A/T)AT to regulate gene expression.

## Downstream target genes of POU-IV are enriched with effector genes likely involved in neural function in the sea anemone

We next sought to identify downstream target genes of POU-IV, based on the criteria that a target gene has at least one POU-IV ChIP peak within the gene locus which includes the promoter region – 350 bp upstream and 100 bp downstream of the transcription start site – and the gene body. Using this criterion, we found a total of 4188 candidate POU-IV downstream target genes (*Supplementary file 1*). We then examined which of these candidate POU-IV target genes were activated/repressed by POU-IV, using publicly available transcriptome data from NvPOU4 mutant polyps and their siblings (*Tournière et al., 2020*). Re-analysis of the transcriptome data identified 577 genes that were downregulated in NvPOU4 mutants relative to their siblings (*Supplementary file 2*), and 657 genes that were upregulated in the mutants (*Supplementary file 3*) (adjusted p-value < 0.01). Consistent with the previous report (*Tournière et al., 2020*), Gene Ontology (GO) terms overrepresented in genes downregulated in NvPOU4 mutants included those related to nervous system function such as 'synaptic transmission' and 'detection of stimulus' (*Supplementary file 4*). GO terms overrepresented in genes upregulated in mutants included 'endoplasmic reticulum', as identified by *Tournière et al., 2020*, as well as a number of additional ones, such as 'proteolysis' and 'activation of signaling protein activity involved in unfolded protein response' (*Supplementary file 5*). Out of the 577 genes downregulated in NvPOU4 mutants relative to their siblings, 293 were POU-IV target genes (*Supplementary file 6*), while out of the 657 genes upregulated in NvPOU4 mutants (*Supplementary file 7*), 178 were POU-IV target genes; we assume that the former represent genes that are directly activated by POU-IV, while the latter represent those directly repressed by POU-IV. Among the POU-IV-repressed genes is the *pou-iv* gene itself, indicating that POU-IV negatively regulates its own expression. GO analysis found that 84 GO terms were overrepresented in the 293 genes directly activated by POU-IV, which include terms related to nervous system function such as 'synaptic transmission' (p-adjusted <0.05) (*Supplementary file 8*). No GO terms were significantly overrepresented in the 178 genes directly repressed by POU-IV (p-adjusted <0.05).

## POU-IV regulates the expression of the hair-cell-specific effector gene *polycystin 1* in the sea anemone

To shed light on the mechanism by which POU-IV regulates hair cell maturation, we assessed which genes were directly activated by POU-IV in hair cells. Among the 577 genes significantly downregulated in NvPOU4 mutants relative to their siblings is a transmembrane receptor-encoding *polycystin 1* (PKD1)-*like* gene (JGI ID: 135278). By using in situ hybridization, we found that this gene was specifically expressed in tentacular epithelial cells whose cell bodies were located in the superficial stratum of the pseudostratified epithelium, resembling the hair cell (*Figure 8A–F*). We discovered by RTPCR that this gene and another *polycystin 1-like* gene (JGI ID: 218539) upstream – which was also one of the 577 genes significantly downregulated in NvPOU4 mutants relative to their siblings – together constitute a single *polycystin 1-like* gene. The transcript of the *polycystin 1-like* is 11,279 bases long and encodes a protein that is 3457 amino acids long (*Figure 8—figure supplement 1*; GenBank accession number, OK338071). ChIP-seq data show that there are two POU-IV-binding motifs around the transcription start site of this locus (*Figure 8G*), suggesting that the *polycystin 1-like* gene is directly regulated by POU-IV.

We predicted the structure of the Polycystin 1-like protein based on sequence similarity to known Polycystin 1 proteins. Transmembrane-spanning regions were predicted by using Phyre2 (*Kelley et al., 2015*) and based on the alignment with human and *Fugu* Polycystin 1 sequences (GenBank accession numbers AAC37576 and AAB86683, respectively). Non-transmembrane-spanning regions were predicted by using NCBI conserved domain search with default Blast search parameters. The *N. vectensis* Polycystin 1-like protein was predicted to have a Polycystin 1-like domain structure, containing the extracellular PKD (polycystic kidney disease) domain and REJ (receptor for egg jelly) module that are uniquely shared by Polycystin 1 proteins (*Moy et al., 1996*; *Bycroft et al., 1999*), the extracellular WSC (cell wall integrity and stress component) carbohydrate-binding domain, the intracellular PLAT (polycystin-1, lipoxygenase, and alpha toxin) domain (*Bateman and Sandford, 1999*), the extracellular TOP (tetragonal opening for polycystins) domain (*Grieben et al., 2017*), and 11 transmembrane domains (*Sandford et al., 1997*; *Figure 8—figure supplement 1*). However, unlike vertebrate Polycystin 1, leucine-rich repeat, C-type lectin, and LDL-like domains that reside in the N-terminal extracellular tail and a coiled-coil domain at the C-terminal intracellular tail were not identifiable. The last six transmembrane domains of Polycystin 1 are thought to be homologous to TRP cation channels including Polycystin 2 (PKD2) (*Mochizuki et al., 1996*); in addition, the TOP domain is shared across Polycystin 1 and 2 (*Grieben et al., 2017*). We therefore generated an amino acid sequence alignment of the TOP domain and transmembrane domains of Polycystin 1 and Polycystin 2 proteins, and used it to carry out maximum likelihood phylogeny estimation. The results robustly placed the newly discovered *N. vectensis polycystin-1-like* within the Polycystin 1 group (*Figure 8—figure supplement 2*). We therefore designate this gene as *N. vectensis polycystin 1*.

To better resolve the cell type in which *polycystin 1* is expressed, we generated a reporter construct using 3704 bp sequence encompassing the two POU-IV-binding motifs and upstream promoter region of the gene (scaffold 353:49,338–53,412). We injected this construct into wildtype zygotes and confirmed reporter gene expression specifically in hair cells at the primary polyp stage (*Figure 8H–K*). In addition, we have validated by in situ hybridization that *polycystin 1* expression is lost in *pou-iv -/-* mutants (n = 5) but not in their siblings (n = 3) (*Figure 8L-S*). Taken together, these results suggest that *polycystin 1* is directly activated by POU-IV in hair cells. To our knowledge, *polycystin 1* represents the first molecular marker specific to cnidarian hair cells.

## POU-IV controls the maturation of hair cells and cnidocytes via distinct gene regulatory mechanisms

Next, we have utilized publicly available single-cell transcriptome data from *N. vectensis* wildtype adults (*Sebé-Pedrós et al., 2018*) to uncover additional candidate genes that are directly activated by POU-IV in hair cells. Both *polycystin-like* gene models (JGI IDs: 135278 and 218539) that are part of the newly discovered *polycystin 1* are uniquely represented in one of Sebe-Pedros et al.'s transcriptomically defined adult cell types ('c79') referred to as the metacells (*Sebé-Pedrós et al., 2018*). We have therefore deduced that the adult metacell c79 represents the hair cell, which enabled identification of additional POU-IV target genes activated in hair cells. Out of the 293 genes directly activated by POU-IV, we found a total of 32 genes that were represented in the adult metacell c79

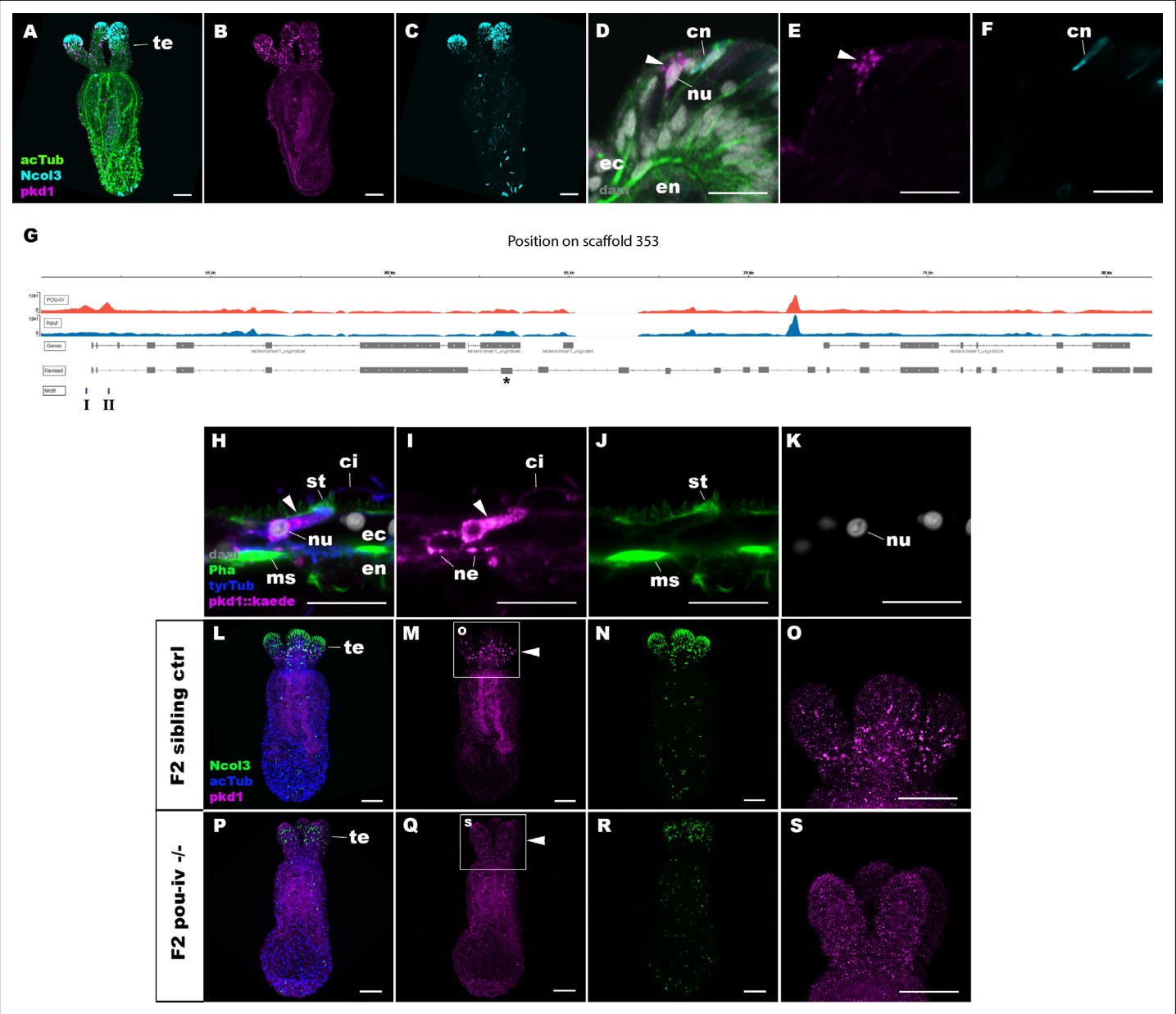

**Figure 8.** POU-IV activates the expression of *polycystin 1* specifically in hair cells. (**A-F**) Confocal sections of primary polyps labeled with an antisense riboprobe against *polycystin 1* transcript ('pkd1') and antibodies against acetylated ∂-tubulin ('acTub') and minicollagen 3 ('Ncol3'; *Zenkert et al., 2011*). Nuclei are labeled with DAPI ('dapi'). **A–C** are side views of the animal with the oral opening facing up. Expression of *polycystin 1* occurs exclusively in the ectoderm of the oral tentacles (te). (**D–F**) Side views of a *polycystin 1*-expressing epithelial cell (arrowhead) in the tentacular ectoderm (ec) with its apical surface facing up. Note that the cell body is localized apically and lacks minicollagen 3 expression. (**G**) A schematic of the *polycystin 1* locus, showing the distribution of POU-IV ChIP DNA ('POU-IV') and input DNA from adult polyps. JGI gene models ('Genes') and the revised gene model based on RTPCR ('Revised') and the locations of the consensus POU-IV-binding motif – AT(A/T)ATT(A/T)AT – are numbered as I and II. X-axis shows the position along the genomic scaffold, and Y-axis shows the number of reads. * shows an exon whose sequence is missing in the publicly available *Nematostella vectensis* genome (v1.0; *Putnam et al., 2007*). (**H–K**) Confocal sections of an oral tentacle of a primary polyp injected with *polycystin 1::kaede* construct, labeled with an antibody against Kaede ('pkd1::kaede'). Filamentous actin is labeled with phalloidin (Pha). The apical surface of the tentacular ectodermal epithelium is to the top. Note that the Kaede-positive cell (arrowhead) has an apical cilium (ci) and stereovilli (st), a central nucleus (nu), and basal neurites (ne), exhibiting morphological hallmarks of a hair cell. No other cell types were found to be Kaede-positive. **L–S**: Confocal sections of a homozygous *pou-iv* mutant ('F2 *pou-iv* -/-', **P-S**) and its sibling control (F2 *pou-iv* +/+ or *pou-iv* +/-, 'F2 sibling ctrl', **L–O**) at the primary polyp stage, labeled with an antisense riboprobe against *polycystin 1* transcript ('pkd1') and antibodies against acetylated ∂-tubulin ('acTub') and minicollagen 3 ('Ncol3'; *Zenkert et al., 2011*). Panels show side views of the animal with the oral opening facing up. Animals lacking mature cnidocysts based on Ncol3 staining were assumed to be *pou-iv* -/- mutants; animals with mature cnidocysts were assumed to be *pou-iv* +/+ or *pou-iv* +/-

*Figure 8 continued on next page*

*Figure 8 continued*

. **O** and **S** are magnified views of tentacles boxed in **M** and **Q**, respectively. Note that cell-type-specific expression of *polycystin 1* in tentacular ectoderm (arrowhead in **M**; **O**) is absent in the POU-IV null mutant (arrowhead in **Q**; **S**), demonstrating that POU-IV is necessary for *polycystin 1* expression. Abbreviations: en, endoderm; cn, cnidocyst; nu, nucleus; ms, muscle fiber. Scale bar: 50 µm (**A–C, L–S**); 10 µm (**D–F, H–K**).

The online version of this article includes the following source data and figure supplement(s) for figure 8:

**Figure supplement 1.** The predicted protein structure of *Nematostella vectensis* Polycystin 1-like.

**Figure supplement 1—source data 1.** Partial cDNA sequence of *Nematostella vectensis polycystin 1-like*, in which the start codon and stop codon are highlighted in green and red, respectively.

**Figure supplement 1—source data 2.** Translated amino acid sequence of *Nematostella vectensis polycystin 1-like*, in which conserved domains are highlighted in light blue for transmembrane-spanning regions and in purple for other domains.

**Figure supplement 2.** *Nematostella vectensis polycystin 1-like* belongs to the Polycystin 1/PKD1 group.

**Figure supplement 2—source data 1.** Alignment of polycystin 1 and polycystin 2 amino acid sequences used to construct phylogeny shown in *Figure 8—figure supplement 2*.

(the presumptive hair cell) (*Supplementary file 9*). They include *potassium channel-like* (NVE ID: 12832; no JGI ID), *GABA receptor-like* (JGI gene ID: 98897), and *polycystin 2* (JGI gene ID: 160849; *Figure 8—figure supplement 2*), in addition to *polycystin 1* identified above. Three of the 32 genes – *coagulation factor/neuropilin-like* (JGI gene ID: 202575), *CD59 glycoprotein-like* (NVE ID: 735; no JGI ID), and *polycystin 1* – are not found in any other metacell. No transcription factor-encoding genes were found to be activated by POU-IV in hair cells. GO analysis found that five GO terms were overrepresented in POU-IV-activated genes in hair cells (p-adjusted <0.05), including 'potassium ion transmembrane transport' and 'sensory perception of sound' (*Supplementary file 10*). In contrast, out of the 178 genes that are repressed by POU-IV, only two genes – *pou-iv* itself and *peptidylglycine alpha-amidating monooxygenase-like* (JGI gene ID: 172604) – are represented in the adult metacell c79 (*Supplementary file 9*). These results are consistent with the hypothesis that POU-IV controls the maturation of hair cells by activating a cell-type-specific combination of effector genes that confer hair cell identity.

We have taken a similar approach to examine how POU-IV regulates the maturation of cnidocytes. Sebé-Pedrós et al. categorized cnidocytes into eight transcriptomically distinct metacells (c1–c8), out of which c8 expresses *pou-iv* (*Sebé-Pedrós et al., 2018*). Of the 293 genes activated by POU-IV, we found three genes in c8, which consisted of two *transmembrane propyl 4-hydroxylase-like* genes (JGI gene IDs: 239358 and 118978) and a *serine transporter-like* gene (JGI gene ID: 238501) (*Supplementary file 11*). *serine transporter-like* and one of the *hydroxylase-like* genes (JGI gene ID: 239358) are represented specifically in cnidocyte metacells and not in others (*Sebé-Pedrós et al., 2018*). As in hair cells, transcription factor-encoding genes were not found to be activated by POU-IV in cnidocytes. Of the 178 genes that are repressed by POU-IV, 13 genes were found in the cnidocyte metacell c8, which included genes encoding zinc finger and Sox transcription factors (*Supplementary file 11*) and showed significant enrichment of the GO term 'transcription from RNA polymerase II promoter' (p-adjusted <0.05; *Supplementary file 12*). Importantly, we found no overlap in genes activated by POU-IV between hair cells and cnidocytes, which indicates that POU-IV controls the maturation of hair cells and cnidocytes by turning on the expression of distinct sets of genes. In addition, POU-IV may have a more significant role as a leaky repressor – to fine-tune gene expression levels – in cnidocytes than in hair cells, as the proportion of POU-IV-repressed genes relative to the total number of POU-IV targets represented in a given metacell was substantially higher in the cnidocyte (13/16; 81.25%) than in the hair cell (3/35; 8.6%). This pattern of gene regulation by POU-IV appears to be specific to cnidocytes. The proportion of POU-IV-repressed genes relative to the total number of POU-IV targets represented in non-cnidocyte, POU-IV-positive adult metacells – namely, c63, c64, c65, c66, c75, c76, and c102, in addition to c79 – was low, ranging from 2.2% to 13%, while that in POU-IV-positive, adult cnidocyte metacells – c100 that expresses spirocyte-specific *minicollagen* (JGI gene ID: 81941) and c101 that expresses nematocyte-specific *minicollagen1* (JGI gene ID: 211803) (*Sebé-Pedrós et al., 2018*) – was 83.9% in c100% and 82.1% in c101 (*Supplementary file 13*), similar to the cnidocyte metacell c8. Taken together, these data suggest that POU-IV directs the maturation of cnidocytes not only by activating a unique combination of effector genes, but also by negatively controlling the expression levels of a larger number of genes, including those encoding transcriptional regulators, in a leaky manner.

Thus, the gene regulatory mechanisms by which POU-IV orchestrates the differentiation of hair cells and cnidocytes appear to be remarkably distinct.

## Discussion

In this paper, we identified the class IV POU homeodomain transcription factor (POU-IV) as an essential developmental regulator of cnidarian mechanosensory hair cells. This is the first discovery of a gene regulatory factor necessary for the development of classical mechanosensory neurons – that transmit mechanosensory information to other cells to elicit coordinated behavior – from a non-bilaterian, early-evolving animal group. Using the starlet sea anemone *N. vectensis,* we have shown that POU-IV is postmitotically expressed in hair cells in the ectodermal epithelium of feeding tentacles during development. In addition, we have found that null mutation of *pou-iv* renders the animal unable to respond to tactile stimulation to its tentacles, and results in the loss of stereovillar rootlets, but not of neurites, in hair cells. Furthermore, we have presented evidence that POU-IV binds to deeply conserved DNA sequence motifs, and directly activates the expression of a unique combination of effector genes, but not transcription factor-encoding genes, specifically in hair cells. Among the POU-IV target effector genes, we discovered the first cnidarian hair cell-specific molecular marker, *polycystin 1,* which encodes a transmembrane receptor of the TRP channel superfamily. The results suggest that POU-IV plays a necessary role in regulating the maturation of mechanosensory hair cells in the sea anemone by directly activating the expression of cell-type-specific effector genes. Our findings strongly support POU-IV being the terminal selector of hair cell identity in the sea anemone.

Several lines of evidence indicate that POU-IV is specifically involved in the maturation, and not progenitor proliferation, initial differentiation, or survival, of hair cells in the tentacular ectoderm in *N. vectensis*. First, POU-IV is postmitotically expressed in hair cells in tentacular ectoderm, and thus is unlikely to have roles in proliferation or generation of their progenitor cells. Second, in POU-IV null mutants, we have found ciliary-cone-bearing epithelial cells that resemble hair cells in morphology and position within the epithelium; these cells are characterized by having an apical cilium surrounded by a circle of microvilli, a pear-shaped cell body located in the superficial stratum of the pseudostratified epithelium, and basal neurites. None of the ciliary-cone-bearing epithelial cells express the pan-cnidocyte marker minicollagen 3, suggesting that the ciliary-cone-bearing cells in POU-IV null mutants do not represent partially differentiated nematocytes. The existence of differentiated hair cells in POU-IV null mutants implies that POU-IV is not involved in the initial differentiation of hair cells. However, the hair-cell-like cells of POU-IV null mutants failed to form a mature apical mechanosensory apparatus with stereovillar rootlets, indicating that POU-IV is essential for maturation of hair cells. Lastly, we have found no evidence that the epithelial cells expressing the non-functional form of POU-IV protein in POU-IV null mutants undergo programmed cell death in the tentacular ectoderm. Thus, POU-IV does not seem to be required for the survival of hair cells in the tentacles. Taken together, these data support the hypothesis that POU-IV regulates the maturation, but not progenitor proliferation, initial differentiation, or survival, of mechanosensory hair cells in the sea anemone.

The loss of stereovillar rootlets in hair cells in *pou-iv* mutants suggests that the POU-IV transcription factor regulates the expression of genes that are involved in stereovillar development. Given that stereovillar rootlets consist of actin filaments, actin-binding proteins may be regarded as potential regulators of stereovillar rootlet formation in hair cells. Among the identified POU-IV target genes expressed in hair cells is *polycystin 1,* which encodes a large transmembrane receptor with multiple extracellular and intracellular domains and TRP-channel-like transmembrane domains. Interestingly, its mouse homolog (PC-1) colocalizes with F-actin in inner ear hair cell stereovilli and is necessary for maintenance of stereovillar structure and normal hearing (*Steigelman et al., 2011*). In addition, PC-1 has been shown to regulate actin cytoskeleton reorganization in canine kidney epithelial cells (*Boca et al., 2007*). If the function of Polycystin 1 in modulating the organization of actin cytoskeleton is broadly conserved, *N. vectensis* Polycystin 1 might control the structural integrity of stereovilli in hair cells through its interaction with F-actin. POU-IV may therefore direct stereovillar development in cnidarian hair cells by activating *polycystin 1*. Functional analysis of *N. vectensis polycystin 1* to evaluate its role in stereovillar development is warranted.

We have proposed that the lack of tentacular response to tactile stimuli in *pou-iv* mutants is due to the loss of structural rigidity in the apical mechanosensory apparatus – stereovilli, in particular – of hair cells, resulting from the failure of hair cells to form stereovillar rootlets. We note, however, that it

could additionally be because of the functional defects in mechanotransduction channels. POU-IV is known to directly activate the expression of the mechanotransduction channel-encoding gene, *mec-4*, that is necessary for touch-cell function in *C. elegans* (*Duggan et al., 1998*). The Polycystin 1 protein discussed above contains transmembrane domains that are homologous to the TRP calcium channel. If this channel is involved in mechanotransduction, the loss of *polycystin 1* expression in *pou-iv* mutants would directly lead to loss of mechanotransduction channel function. This hypothesis may be evaluated by specifically examining the role of the channel-encoding region of *N. vectensis polycystin 1* in mechanotransduction.

Alternatively, the loss of *polycystin 1* expression may indirectly perturb channel function. In epithelial cells of vertebrate kidneys, Polycystin 1 interacts with the calcium ion channel Polycystin 2 to form a complex that functions as a fluid flow sensor with Polycystin 1 acting as a mechanosensitive cell surface receptor and Polycystin 2 as an ion-permeant pore (reviewed in *Delmas, 2004*). The *N. vectensis* genome encodes *polycystin 2* (*Figure 8—figure supplement 2*), which is co-expressed with *polycystin 1* in the adult metacell c79 (i.e. the hair cell) (*Supplementary file 9*; *Sebé-Pedrós et al., 2018*). If these two proteins form a mechanically gated ion channel complex in hair cells as in vertebrate kidney epithelial cells, the loss of expression of *polycystin 1* would perturb the ability of the complex to sense mechanical stimuli, resulting in defects in channel function. To explore this hypothesis, the important next step will be to assess whether Polycystin 1 and 2 form a complex in *N. vectensis*.

We note that, although our findings are consistent with the hypothesis that cnidarian hair cells function as mechanosensors, we do not rule out the possibility that cnidarian hair cells might be multimodal sensory cells; they might have additional functions as chemoreceptors and/or photoreceptors. Indeed, hair cells of the sea anemone *Haliplanella luciae* have been reported to respond to N-acetylated sugars by elongating their stereovilli (*Mire-Thibodeaux and Watson, 1994*, *Watson and Roberts, 1995*), indicative of chemosensory function. The metacell c79 (the presumptive hair cell) of adult *N. vectensis* expresses several G-protein-coupled receptor (GPCR)-encoding genes (*Supplementary file 14*; *Sebé-Pedrós et al., 2018*), some of which might encode chemosensory receptors; none were found to encode opsins. Functional analyses of these GPCRs may shed light on the molecular basis of sensory multimodality in cnidarian hair cells.

Our results indicate that the role for POU-IV in mechanoreceptor development is broadly conserved across Cnidaria and Bilateria. As mentioned above, one of the vertebrate *pou-iv* paralogs (Brn3c) is necessary for maturation and survival of inner ear hair cells in mice (*Xiang et al., 1998*; *Xiang et al., 1997a*, *Erkman et al., 1996*). Likewise, in *C. elegans*, differentiation of mechanosensory touch cells requires a *pou-iv* ortholog *unc-86* (*Chalfie and Sulston, 1981*; *Chalfie and Au, 1989*; *Finney and Ruvkun, 1990*; *Duggan et al., 1998*; *Chalfie et al., 1981*). In Cnidaria, *pou-iv* is expressed in mechanosensory organs in scyphozoan and hydrozoan jellyfish (*Nakanishi et al., 2010*; *Hroudova et al., 2012*), and is necessary for differentiation of the lineage-specific mechanosensory-effector cell, the cnidocyte, in *N. vectensis* (*Tournière et al., 2020*). In this report, we have demonstrated that *pou-iv* has an essential role in the maturation of the classical mechanosensory neuron of Cnidaria – the concentric hair cell – using *N. vectensis*. These comparative data show that POU-IV-dependent regulation of mechanosensory cell differentiation is pervasive across Cnidaria and Bilateria, and likely predates their divergence. How early the role of POU-IV in mechanoreceptor differentiation emerged in animal evolution remains unresolved, and requires comparative data from placozoans and sponges, which are wanting.

We note, however, that POU-IV has a broad role in the differentiation of multiple neural cell types across Cnidaria and Bilateria. In *N. vectensis*, POU-IV expression is not restricted to mechanosensory hair cells and cnidocytes, but also found in RFamidergic neurons and *NvElav1*-positive endodermal neurons (*Tournière et al., 2020*). Likewise in Bilateria, POU-IV regulates the differentiation of a variety of neural cell types beyond mechanosensory cells, including chemosensory neurons in insects (*Clyne et al., 1999*) and photosensory neurons in vertebrates (retinal ganglion cells; e.g. *Erkman et al., 1996*; *Gan et al., 1996*). Therefore, it seems plausible that POU-IV was ancestrally involved in the differentiation of multiple neural cell types in addition to mechanosensory cells.

Interestingly, POU-IV is required for normal development of stereovilli in hair cells in both sea anemones (this study) and mice (*Xiang et al., 1998*), raising the possibility that POU-IV controlled the formation of the apical sensory apparatus of mechanosensory cells in the last common ancestor

of Cnidaria and Bilateria, potentially via regulation of *polycyctin 1*. Alternatively, the essential role for POU-IV in stereovillar formation in mechanosensory cells could have evolved independently in Cnidaria and vertebrates. Comparative studies of the mechanism of stereovillar formation across sea anemones and vertebrates, along with mechanistic studies of POU-IV gene function in phylogenetically informative taxa, such as medusozoan cnidarians and acoel bilaterians, are needed to evaluate these alternative hypotheses.

Regulatory factors acting upstream of POU-IV in cnidarian hair cell development remain unknown. In Bilateria, members of the Atonal basic-loop-helix-loop-helix (bHLH) transcription factor family appear to have a conserved role in positive regulation of POU-IV expression (reviewed in *Leyva-Díaz et al., 2020*). For instance, vertebrate *atonal* genes act upstream of *pou-iv* genes to drive the differentiation of inner ear hair cells (*Ikeda et al., 2015*; *Masuda et al., 2011*; *Yu et al., 2021*) and retinal ganglion cells (*Liu et al., 2001*). Similarly, *C. elegans atonal* ortholog *lin-32* controls the expression of the *pou-iv* ortholog *unc-86* in touch sensory neurons (*Baumeister et al., 1996*). Although cnidarians lack unambiguous *atonal* orthologs, they have divergent bHLH genes that belong to the Atonal superfamily, which consists of Atonal and related bHLH gene families including Neurogenin, and NeuroD (*Gyoja et al., 2012*; *Simionato et al., 2007*). Whether these *atonal-like* bHLH factors function upstream of POU-IV in the context of cnidarian hair cell development needs to be assessed, as it may provide insights into the evolution of gene regulatory mechanisms underpinning mechanoreceptor development across Cnidaria and Bilateria.

In light of new comparative data reported herein emerges a model of mechanosensory cell differentiation in the last common ancestor of Cnidaria and Bilateria. We assume that the embryo of the Cnidaria-Bilateria ancestor had neurogenic ectoderm (*Nakanishi et al., 2012*). During late embryogenesis or postembryonic development of this ancestor, mechanosensory cell progenitors differentiated into postmitotic sensory cells in the ectoderm, extending apical cilia and basal neurites. These postmitotic sensory cells expressed the terminal selector POU-IV, which translocated to the cell nuclei, and bound to the DNA recognition motif GCAT(A/T)ATT(A/T)AT (i.e. the consensus motif across Bilateria and Cnidaria) associated with target genes in these cells. This activated the expression of effector genes, possibly including *polycystin 1,* whose protein products generated mechanoreceptor-specific structures necessary for mechanosensory function, such as the apical mechanosensory apparatus consisting of a cilium surrounded by a ring of stereovilli. The mature identity of the mechanosensory cell was thereby established.

Following the divergence of Cnidaria and Bilateria, POU-IV may have been recruited for the evolution of the lineage-specific mechanosensory effector – the cnidocyte – in Cnidaria, as *pou-iv* is essential for cnidocyte development in *N. vectensis* (*Tournière et al., 2020*). How POU-IV would have come to direct cnidocyte development remains unclear. Given that both hair cells and cnidocytes are mechanosensory cell types, it seems reasonable to expect that an ancestral POU-IV gene regulatory network that defined mechanosensory cell identity was repurposed for the emergence of cnidocytes, and should be shared across these two cell types. However, we found no evidence in support of this hypothesis; instead, our results suggest that POU-IV turns on distinct sets of genes in each cell type. One possible evolutionary scenario to account for this observation is that POU-IV initially activated the same battery of effector genes in the ancestral cnidocytes and hair cells, but POU-IV target genes diverged substantially during cnidarian evolution so that they no longer overlap between the two cell types. Another possibility is that POU-IV regulated a unique set of genes in the ancestral cnidocytes when POU-IV became part of the cnidocyte developmental gene regulatory network. This possibility seems conceivable if POU-IV expression was activated in epigenetically distinct cell lineages, so that between the cnidocyte lineage and the hair cell lineage (1) POU-IV cooperated with different co-factors, and/or (2) the accessibility of POU-IV target genes differed, which would result in differential expression of POU-IV target genes. Evidence from bilaterian models such as *C. elegans* and mice indicates that POU-IV cooperates with a range of different co-factors to define distinct neural identities (reviewed in *Leyva-Díaz et al., 2020*), suggesting an important role for POU-IV co-factors in the diversification of neural cell types. Whether evolution of POU-IV co-factors played a role in the evolution of cnidocytes remains to be tested. Investigation into the mechanism by which POU-IV activates distinct sets of genes across cnidocytes and hair cells will be the critical next step for shedding light on how POU-IV may have contributed to the evolution of the novel mechanosensory cell type of Cnidaria.

## Materials and methods

**Key resources table**

| Reagent type (species) or resource | Designation | Source or reference | Identifiers | Additional information |
|---|---|---|---|---|
| Strain, strain background (*Nematostella vectensis*) | F1 *pou-iv* +/- | This paper | Nv F1 *pou-iv* +/-, this paper | Maintained in N. Nakanishi Lab, University of Arkansas |
| Antibody | Anti- *Nematostella vectensis* POU-IV/Brn-3; rabbit; polyclonal | This paper | RRID:AB_2895562 | IF (1:200); stored in N. Nakanishi Lab, University of Arkansas |

### Animal culture

*N. vectensis* were cultured as previously described (*Fritzenwanker and Technau, 2002*; *Hand and Uhlinger, 1992*).

### RNA extraction, cDNA synthesis, and gene cloning

Total RNA was extracted from a mixture of planulae and primary polyps using TRIzol (Thermo Fisher Scientific). cDNAs were synthesized using the SMARTer RACE cDNA Amplification Kit (Cat. No. 634858; Takara, Mountain View, CA). In silico predicted *pou-iv* gene sequence was retrieved from the Joint Genome Institute genome database (*N. vectensis* v1.0, protein ID 160868; http://genome.jgi-psf.org/Nemve1/Nemve1.home.html). 5′ and 3′ RACE PCR experiments were conducted, following manufacturer's recommendations, in order to confirm gene prediction. Gene-specific primer sequences used for RACE PCR are: 3′ RACE Forward 5′-CGATGTCGGGTCCGCGCTTGCACATTTG-3′; 5′ RACE Forward 5′-GCCGCGCCGATAGACGTGCGTTTACG-3′. RACE PCR fragments were cloned into a pCR4-TOPO TA vector using the TOPO TA Cloning kit (Cat. No. K457501; Thermo Fisher Scientific), and sequenced at Genewiz, NJ.

The *polycystin 1* cDNA sequence (GenBank accession number: OK338071) was obtained by subcloning small overlapping gene fragments (1.5–4 kb). Gene fragments were generated by RTPCR using RACE-ready cDNAs as templates. Gene-specific primer sequences used to amplify *polycystin 1* cDNA are listed in *Supplementary file 15*. The PCR products were then cloned into a pCR4-TOPO TA vector using the TOPO TA Cloning kit (Cat. No. K457501; Thermo Fisher Scientific), and sequenced at Eurofins Genomics, KY.

### Generation of an antibody against *N. vectensis* POU-IV

An antibody against a synthetic peptide CQPTVSESQFDKPFETPSPINamide corresponding in amino acid sequence to N-terminal QPTVSESQFDKPFETPSPIN of *N. vectensis* POU-IV (*Figure 4A*) was generated in rabbit (YenZym Antibodies, LLC). TBLASTN search of the antigen sequence against the *N. vectensis* genome (http://metazoa.ensembl.org/Nematostella_vectensis/Info/Index) yielded a single hit at the *pou-iv* locus (NEMVEscaffold_16:1069268–1069327); there were no significant matches to other loci. Following immunization, the resultant antiserum was affinity purified with the CQPTVSESQFDKPFETPSPINamide peptide.

### CRISPR-Cas9-mediated mutagenesis

Twenty-nt-long sgRNA target sites were manually identified in the *N. vectensis pou-iv* genomic locus. To minimize off-target effects, target sites that had 17 bp-or-higher sequence identity elsewhere in the genome (*N. vectensis v1.0*; http://genome.jgi.doe.gov/Nemve1/Nemve1.home.html) were excluded. Selected target sequences were as follows.

    5′- CTACGATGCGCACGATATTT-3′ (Cr1)
    5′- ACGAGAGCTGGAATGGTTCG-3′ (Cr2)
    5′- TAAACGCACGTCTATCGGCG-3′ (Cr3)
    5′- AATAATGGACATCTACGCCG-3′ (Cr4)

The sgRNA species were synthesized in vitro (Synthego) and mixed at equal concentrations. The sgRNA mix and Cas9 endonuclease (PNA Bio, PC15111, Thousand Oaks, CA) were co-injected into fertilized eggs at concentrations of 500 and 1000 ng/μl, respectively.

### Genotyping of embryos and polyps

Genomic DNA from single embryos or from tentacles of single polyps was extracted by using published protocols (*Ikmi et al., 2014*; *Silva and Nakanishi, 2019*), and the targeted genomic locus

was amplified by nested PCR. Primer sequences used for nested genomic PCR are: '1' Forward 5'-CGAATTCCTCTGCAATAATCACTGATCG-3', '1' Reverse 5'-CTCGTTGGCAGGTGCGGAAAGAG-3', '2' Forward 5'-CGTTCGACTTCATTTCCGCTCGTC-3', '2' Reverse 5'-CGGAAGTTAACGTCGTTAAT GCGAAGG-3'. To determine the sequence of mutant alleles, PCR products from genomic DNA extracted from F1 mutant polyps were gel-purified, cloned, and sequenced by using a standard procedure. Using the sequence information of the *pou-iv-* mutant allele, genotyping primers for F2 animals were designed as follows (*Figure 4B*).

Forward 5'- CGTTCGACTTCATTTCCGCTCGTC-3'

Reverse (1), 5'- GCCGCGCCGATAGACGTGCGTTTACG-3' (*pou-iv+* -specific; expected size of PCR product, 689 bp)

Reverse (2), 5'- CGGAAGTTAACGTCGTTAATGCGAAGG-3' (expected size of PCR product: *pou-iv+*, 1312 bp; *pou-iv-*, 558 bp)

## Behavioral analysis

Animals were selected for behavior analyses if they were 10–16 dpf, unfed, had reached the primary polyp stage, and had two or more tentacles present. Animals were only tested if their tentacles protruded from their bodies at time of testing initiation. All behavior experiments were performed with the experimenter blind to the animal's genotype until after testing was completed. Animals were allowed to rest for at least 2 hr between tests. Behavioral analyses were performed under a Zeiss Stemi 508 microscope with Nikon DSL-4 camera attachment.

To examine response to touch, a hair attached to a microdissection needle holder (Roboz Surgical Instrument Co., Gaithersburg, MD) was pressed briefly on the distal end of each tentacle. The stimulus was applied once more to remaining unretracted tentacles, to ensure that a tentacle was not missed during the first stimulus. The number of primary polyps that retracted one or more tentacles upon touch was counted. If any other part of the body was touched accidentally during tentacle stimulation, data for that animal were discarded and the trial was repeated 2 hr or more after the previous test.

Chemosensory response of primary polyps to *Artemia* chemical cues was analyzed as follows. *Artemia* shrimp extract was made from 1-day-old *Artemia* brine shrimp, ground with a micropestle (USA Scientific) in 1/3 artificial seawater (Instant Ocean), at a concentration of approximately one shrimp per 1 μl. Two μl of shrimp extract was applied with an Eppendorf pipette to the head and tentacle area of each sea anemone. The animal was observed for 1 min to examine the occurrence of tentacular retraction.

## CM-DiI labeling

The lipophilic tracer CM-DiI (Molecular Probes, C7000) was used to label the cell membrane of a subset of mature hair cells of the polyp tentacles. Primary polyps were incubated in 1/3 seawater with 10 μM CM-DiI for 1 hr at room temperature (RT). The labeled polyps were rinsed in fresh 1/3 seawater and were anesthetized in 2.43% MgCl$_2$ for 20 min. They were then fixed in 4% formaldehyde for 1 hr at RT. Specimens were washed in PBSTr (1×PBS + 0.5% Triton-X100) for 1 hr to permeabilize the tissue, before labeling filamentous actin and nuclei with AlexaFluor 488-conjugated phalloidin (1:25, Molecular Probes A12379) and the fluorescent dye 4',6-diamidino-2-phenylindole (DAPI; 1:1000, Molecular Probes D1306), respectively.

## EdU pulse labeling

Tentacle-bud-stage animals and primary polyps were incubated in 1/3 seawater containing 200 μM of the thymidine analogue, EdU (Click-iT EdU AlexaFluor 488 cell proliferation kit, C10337, Molecular Probes), for 20 min to label S-phase nuclei. Following washes in fresh 1/3 seawater, the animals were immediately fixed as described previously (*Martindale et al., 2004*; *Nakanishi et al., 2012*), and immunohistochemistry was then carried out as described below. Following the immunohistochemistry procedure, fluorescent labeling of incorporated EdU was conducted according to the manufacturer's recommendations prior to DAPI labeling.

## Western blotting

Three- to four-week-old polyps were lysed in AT buffer (20 mM HEPES pH 7.6, 16.8 mM Na$_4$P$_2$O$_7$, 10 mM NaF, 1 mM Na$_3$VO$_4$, 0.5 mM DTT, 0.5 mM EDTA, 0.5 mM EGTA, 20% glycerol, 1% Triton

X-100, and protease inhibitor cocktail [Sigma]) on ice with a plastic pestle in a microcentrifuge tube until there were no large fragments. The mixture was then sonicated with a Branson Digital Sonifier three times with the setting of 0.5 s on 1 s off for 10 s at an amplitude of 10%. NaCl was added to the lysate to a final concentration of 150 mM. The samples were centrifuged at 21,000 $g$ for 20 min at 4°C. The supernatant was transferred to a new centrifuge tube and the pellet was discarded. Protein concentration of the supernatant was determined by Bradford Reagent (Sigma). The proteins were then separated on a 12% SDS-PAGE (40 µg protein/lane), transferred to a PVDF membrane (Amersham Hybond; 0.2 µm). After blocking with the Odyssey Blocking Buffer (TBS) for 30 min at RT, the membrane was incubated with an anti-POU-IV polyclonal antibody (rabbit, 1:1000) at 4°C overnight. The membranes were then washed extensively with TBST and incubated with 1:10,000 IRDye 800CW donkey anti-rabbit IgG at RT for 1 hr. After washing, protein bands were visualized on a LI-COR (9120) Imaging System. Anti-tubulin (T6793 Sigma) was used as a loading control.

## Chromatin immunoprecipitation sequencing

Adult animals (~1.2 g wet weight) were harvested and washed with PBS twice. The animals were treated with 2% formaldehyde in PBS for 12 min at RT, and the cross-linking reagent was quenched with 0.125 M glycine for 5 min at RT. After washing with PBS twice, the cross-linked samples were resuspended in 10 ml buffer1 (50 mM HEPES, pH 7.5, 140 mM NaCl, 1 mM EDTA, 10% glycerol, 0.5% NP-40, 0.25% Triton X-100, 1 mM DTT, and protease inhibitors [Sigma]) and lysed with 10 slow strokes of a tight-fitting pestle (type B) in a 15 ml *Dounce* homogenizer. The lysate was centrifuged at 500 $g$ for 5 min at 4°C, and the resulting pellet was resuspended in 10 ml buffer1 and homogenized as described above. The homogenization processes were repeated one to two more times. In the last homogenization, the lysate was centrifuged at 2000 $g$ for 10 min at 4°C, and the pellet (nuclei) was resuspended in 4 ml SDS lysis buffer (50 mM Tris-HCl, pH 8.0, 10 mM EDTA, 1% SDS, and protease inhibitors). The chromatin was sheared to 200–500 bp fragments by sonicating the samples 12 times (1" on and 1" off for 1 min) at an amplitude of 50% with a Branson Digital Sonifier. The sonicated samples were centrifuged at 21,000 $g$ for 10 min at 4°C and then diluted 10× with CHIP dilution buffer (17.7 mM Tris-HCl, pH 8.0, 167 mM NaCl, 1.2 mM EDTA, 1.1% Triton X-100, 0.01% SDS, and protease inhibitors). After the lysate was cleared with Protein A and G magnetic beads (Cell Signaling), 50 µl of the cleared sample was set aside as input DNA, and 5 ml of lysate was incubated with 10 µg anti-Brn3 rabbit polyclonal antibody, which was conjugated to 30 µl of protein A + G magnetic beads. After incubation at 4°C overnight, the beads were washed three times with 1 ml of low salt buffer (20 mM Tris-HCl, pH 8.0, 150 mM NaCl, 2 mM EDTA, 1% Triton X-100, 0.1% SDS), three times with 1 ml of high salt buffer (20 mM Tris-HCl, pH 8.0, 500 mM NaCl, 2 mM EDTA, 1% Triton X-100, 0.01% SDS), three times with 1 ml of LiCl wash buffer (10 mM Tris-HCl, pH 8.0, 0.25 M LiCl, 0.5% NP-40, 0.5% sodium dexycholate, and 1 mM EDTA), and three times with 1 ml of TE buffer (10 mM Tris-HCl, pH 8.0, and 1 mM EDTA). The chromatin was eluted in SDS elution buffer (50 mM Tris-HCl, pH 8.0, 1% SDS, and 1 mM EDTA), followed by reverse cross-linking at 65°C overnight. After being treated with RNase A (1 mg/ml) at 37°C for 1 hr and then with protease K (0.2 mg/ml) at 45°C for 1 hr, the DNA fragments were purified with QIAquick Spin columns (QIAGEN) and the purified DNA samples were quantified by Qubit4 (Thermo Fisher). 20 ng of the immunoprecipitated DNA or input DNA was used to generate a library with the NEBNext Ultra II DNA Library kit following the manufacturer's protocol. Libraries were initially quantified by Qubit4 and the size profiles were determined by TapeStation (Agilent) and then quantified by qPCR (NEBNext Library Quant Kit) for high-throughput sequencing. Four biological replicates of libraries of immunoprecipitated DNA and the input DNA were pooled in equimolar ratio and the pooled libraries were sequenced on a DNBseq Sequencing platform (BGI, China) for PE 100 bp.

## Expression and purification of POU-IV protein

cDNA-encoding POU-IV was inserted into a modified PET28a plasmid in which POU-IV was expressed under a 2× Flag tag and a tobacco etch virus protease cleavage site by PCR using forward primer 5'-GATGACAAGGGAGGTGGATCCATGAACCGGGACGGATTGCTTAAC-3' and reverse primer 5'-GGTGGTGGTGGTGCTCGAGTCAATGTACGGAGAACTTCATTCTC-3'. The construct was transformed into BL21 (DE3) cells (C2530, NEB). After the transformed cells were grown in LB medium to 0.6 at OD600, the expression of the protein was induced by 1 mM of isopropyl β-ᴅ-1-thiogalactopyranoside at 30°C

for 5 hr. The cells were lysed by sonication in lysis buffer (20 mM Tris pH 7.5, 150 mM NaCl, 1% Triton X-100, 10% glycerol, 1 mM EDTA, and protease inhibitor [P8340, Sigma]). The lysate was cleared by centrifugation at 30,000 $g$ for 30 min at 4°C, and the supernatant was incubated with anti-Flag M2 Affinity Gel (A2220, Sigma) overnight at 4°C. After washing with wash buffer (20 mM Tris pH 7.5, 150 mM NaCl, 0.5% Triton X-100, and 1 mM EDTA), the bound proteins were eluted with elution buffer (50 mM Tris pH 7.5, 30 mM NaCl, and 0.25 mg/ml 3× Flag peptide [F4799, Sigma]). The buffer for the eluted protein was changed to 20 mM Tris pH 7.5, and 100 mM NaCl using an Amicon Ultra Centricon with 10 kDa cut-off. The purified protein was stored at –80°C for further use.

## Electrophoretic mobility shift assay

The biotin-labeled DNA oligonucleotides with or without motif were purchased from Integrated DNA Technologies. For the experiment shown in *Figure 7C*, the probe sequence with motif was 5'-AAACAAAGATTCTAAGCATC***CATTATTAAT***ATACATCCCTAGAAAAAATC-3' (motif in bold and italics; scaffold 353:52091–52140, https://mycocosm.jgi.doe.gov/Nemve1/Nemve1.home.html), and that without motif was 5'- ATCGAAAACAAAGATTCTAAGCATCCATACATCCCTAGAAAAAATCTCCGC-3'.

The two complementary strands were annealed together by mixing equivalent molar amounts of each oligonucleotide, heating at 95°C for 5 min, and slow cooling on bench to RT. Gel mobility shift assay was carried out using Gelshift Chemiluminescent EMSA kit (#37341, Active Motif) with modifications. Briefly, 0.25 µg POU-IV protein, 20 fmol biotin-labeled probes with or without motif were incubated in binding buffer (10 mM Tris pH 7.4, 50 mM KCl, 2 mM MgCl$_2$, 1 mM EDTA, 1 mM DTT, 5% glycerol, 4 µg/ml BSA, and 0.125 µg/µl salmon sperm DNA) in a total volume of 20 µl at RT for 30 min. For the competition, unlabeled probe was added to the reaction mixture at 300-fold molar excess of the biotin-labeled probe. For supershift assay, 2 µg POU-IV antibody was incubated with POU-IV protein for 1 hr at RT before the biotin-labeled probe was added. The DNA-protein complexes were separated with a 5% nondenaturing polyacrylamide gel in 0.5× TBE buffer at 100 V for 1 hr. The probes were then transferred to a positively charged nylon membrane (Nytran SPC, Cytiva) in 0.5× TBE buffer at 380 mA for 30 min at 4C°. After cross-linking the transferred probes to the membrane by CL-1000 Ultraviolet Crosslinker (UVP) at 120 mJ/cm$^2$ for 1 min, the membrane was incubated with HRP-conjugated streptavidin, and the chemiluminescence of the biotin-labeled probes was detected with ECL HRP substrate in X-ray film.

## Immunofluorescence, in situ hybridization, and TUNEL

Animal fixation and immunohistochemistry were performed as previously described (*Martindale et al., 2004*; *Nakanishi et al., 2012*). For immunohistochemistry, we used primary antibodies against POU-IV (rabbit, 1:200), minicollagen 3 (guinea pig, 1:200; *Zenkert et al., 2011*), Kaede (rabbit; 1:500; Medical & Biological Laboratories, PM012M), and tyrosinated ∂-tubulin (mouse, 1:500, Sigma T9028), and secondary antibodies conjugated to AlexaFluor 568 (1:200, Molecular Probes A-11031 [anti-mouse] or A-11036 [anti-rabbit]) or AlexaFluor 647 (1:200, Molecular Probes A-21236 [anti-mouse] or A-21245 [anti-rabbit]). Nuclei were labeled using the fluorescent dye DAPI (1:1000, Molecular Probes D1306), and filamentous actin was labeled using AlexaFluor 488-conjugated phalloidin (1:25, Molecular Probes A12379). For in situ hybridization, antisense digoxigenin-labeled riboprobes against *N. vectensis pou-iv* and *polycystin 1* were synthesized from 5' and 3' RACE products, respectively (MEGAscript transcription kit; Ambion), and were used at a final concentration of 1 ng/µl. TUNEL assay was carried out after immunostaining, by using In Situ Cell Death Detection Kit (TMR red Cat. No. 1684795, Roche, Indianapolis, IN) according to manufacturer's recommendation. Specimens were mounted in Prolong-Gold antifade reagent (Molecular Probes, P36930). Fluorescent images were recorded using a Leica SP5 Confocal Microscope or a Zeiss LSM900. Images were viewed using ImageJ.

## Transmission electron microscopy

One- to four-week old primary polyps were anesthetized in 2.43% MgCl$_2$ for 20 min, and then fixed in 2.5% glutaraldehyde and 0.1 M cacodylate buffer at 4°C overnight. Fixed polyps were washed four times in 0.1 M cacodylate buffer for 10 min, and post-fixed for 1 hr in 0.1 M cacodylate buffer and 1% OsO$_4$. Specimens were rinsed with five 5 min washes of 0.1 M cacodylate buffer, followed by dehydration in a graded ethanol series consisting of 15 min washes in 30%, 50%, 70%, 80%, and 95% ethanol, followed by two 15 min washes in 100% ethanol. Dehydrated polyps were placed in a 1:1 solution of

ethanol/Spurr's resin and left in a desiccator for 1 hr. The ethanol/resin mixture was replaced with a 100% resin solution, and polyps were left in a desiccator overnight. Samples were then transferred to flat-embedding molds filled with 100% Spurr's resin and placed in an oven at 70°C for 14 hr.

Blocks containing embedded polyps were trimmed with a razor blade and cut into ultra-thin sections using a diamond knife on a Sorval Porter-Blum ultramicrotome. Sections were transferred to carbon/formvar-coated copper grids, which were then stained with 2% uranyl acetate and lead citrate and viewed on a JEOL JEM-1011 transmission electron microscope at 100 kV.

## Generation of *kaede* transgenic animals

The *pou-iv::kaede* and *pkd1::kaede* transgenic animals were produced by I-SceI-mediated transgenesis as described previously (*Renfer et al., 2010*) with modifications. To generate *pou-iv::kaede* plasmid, 3199 bp genomic sequence upstream of the start codon of the *N. vectensis pou-iv* (scaffold 16: 1065408–1068606; https://mycocosm.jgi.doe.gov/Nemve1/Nemve1.home.html) was cloned in front of the open reading frame of the Kaede gene (*Ando et al., 2002*) by FastCloning (*Li et al., 2011*). To generate *pkd1::kaede* plasmid, 3704 bp genomic sequence upstream of the 5th base in exon 3 of the *N. vectensis polycystin 1* (scaffold 353: 49524–53227; https://mycocosm.jgi.doe.gov/Nemve1/Nemve1.home.html) was cloned in front of the open reading frame of the Kaede gene. The plasmid was digested with I-SceI for 15–30 min at 37°C and injected into zygotes at 50 ng/μl. The injected animals were raised to primary polyps, and Kaede was visualized by using an anti-Kaede antibody.

## Phylogenetic analysis

Sequence alignment and phylogenetic analyses were performed on the Geneious Prime platform (v2019.2.). *polycystin 1* and *polycystin 2* sequences were retrieved from GenBank at the NCBI website (http://blast.ncbi.nlm.nih.gov/Blast.cgi), either directly or via the protein BLAST search using the *N. vectensis* sequences as queries. Peptide sequences were aligned with MUSCLE (v3.7) (*Edgar, 2004*) configured for highest accuracy (MUSCLE with default settings). After alignment, ambiguous regions (i.e. containing gaps and/or poorly aligned) were manually removed. The final alignment spanned the conserved TOP and transmembrane domains over 323 amino acid sites (*Figure 8—figure supplement 2—source data 1*). Phylogenetic trees were reconstructed using the maximum likelihood method implemented in the PhyML program (*Guindon and Gascuel, 2003*). The WAG substitution model (*Whelan and Goldman, 2001*) was selected assuming an estimated proportion of invariant sites and four gamma-distributed rate categories to account for rate heterogeneity across sites. The gamma shape parameter was estimated directly from the data. Reliability for internal branches of maximum likelihood trees was assessed using the bootstrapping method (100 bootstrap replicates).

## RNA-seq data analysis

The accession number from the RNA-seq data used in this study is E-MTAB-8658. The raw fastq files were filtered for low-quality reads using Trimmomatic v0.39 (SLIDINGWINDOW:4:15, MINLEN:60, HEADCROP:10) (*Bolger et al., 2014*). Filtered reads were aligned to the *N. vectensis* genome (ENA accession: GCA_000209225) using STAR v 2.7.5 a (sjdbOverhang: 99) (*Dobin et al., 2013*). The alignment files were processed using Samtools v1.9 (*Danecek et al., 2021*) and reads on genes were counted using HTSeq v0.12.4 (-t gene) (*Anders et al., 2015*). Genome annotation reported by *Fredman et al., 2013*, was used. The differential expression analysis and normalization were performed in R, using the DESeq2 (*Love et al., 2014*) and Apeglm (*Zhu et al., 2019*) packages.

## ChIP-seq data analysis

ChIP-seq data are available at the BioProject database (accession number: PRJNA767103). The raw data were trimmed using Trimmomatic v0.39 (*Bolger et al., 2014*). Reads were aligned to the *N. vectensis* genome (ENA accession: GCA_000209225) using STAR v2.7.5a (*Dobin et al., 2013*) and alignment files were processed using Samtools v1.9 (*Danecek et al., 2021*). Peak calling was performed in the aligned readings with MACS2 (*Zhang et al., 2008*). The quality of the peaks in the replicates (n = 3) was checked using phantompeakqualtools (*Landt et al., 2012*). To improve sensitivity and specificity of peak calling, and identify consensus regions of the multiple replicates, we used Multiple Sample Peak Calling (MSPC: -r tec -w 1e-4 -s 1e-8) (*Jalili et al., 2015*). A de novo motif search

and motif enrichment was performed using RSAT local-word-analysis (*Thomas-Chollier et al., 2012*). The motif comparison tool TomTom (*Gupta et al., 2007*) was used to search enriched motifs against the Jaspar database (*Fornes et al., 2020*).

The scripts for RNA-seq and ChIP-seq analysis are available here: https://github.com/pyrosilesl97/POU-IV_analysis (copy archived at swh:1:rev:463178242d112edab7094c12e093dd780177885b, *Loeza, 2022*).

## Acknowledgements

We thank Suat Özbek for providing us with the antibodies against minicollagens, and Sakura Rieck for helping with the behavioral analysis of *pou-iv* mutants. We are also grateful to Betty Martin and Mourad Benamara at the Arkansas Nano & Bio Materials Characterization Facility for their assistance with confocal microscopy and transmission electron microscopy. Finally, we would like to thank reviewers for comments on the earlier version of the manuscript, which greatly improved the manuscript.

## Additional information

### Funding

| Funder | Grant reference number | Author |
| --- | --- | --- |
| National Science Foundation | 1931154 | Nagayasu Nakanishi |

The funders had no role in study design, data collection and interpretation, or the decision to submit the work for publication.

### Author contributions

Ethan Ozment, Investigation, Methodology, Validation, Writing - original draft, Writing - review and editing; Arianna N Tamvacakis, Formal analysis, Investigation, Methodology, Writing - original draft; Jianhong Zhou, Investigation, Methodology, Validation, Writing - original draft; Pablo Yamild Rosiles-Loeza, Formal analysis, Investigation, Software, Visualization, Writing - original draft; Esteban Elías Escobar-Hernandez, Formal analysis, Investigation, Software; Selene L Fernandez-Valverde, Data curation, Resources, Software, Supervision, Validation, Writing - review and editing; Nagayasu Nakanishi, Conceptualization, Data curation, Formal analysis, Funding acquisition, Investigation, Methodology, Project administration, Resources, Supervision, Validation, Visualization, Writing - original draft, Writing - review and editing

### Author ORCIDs

Pablo Yamild Rosiles-Loeza ⓘ http://orcid.org/0000-0002-4569-9076
Esteban Elías Escobar-Hernandez ⓘ http://orcid.org/0000-0002-8892-5141
Nagayasu Nakanishi ⓘ http://orcid.org/0000-0001-7516-5078

### Decision letter and Author response

Decision letter https://doi.org/10.7554/eLife.74336.sa1
Author response https://doi.org/10.7554/eLife.74336.sa2

## Additional files

### Supplementary files

• Supplementary file 1. List of 4188 candidate POU-IV downstream target genes.

• Supplementary file 2. List of 577 genes significantly downregulated in NvPOU4 mutants relative to their siblings.

• Supplementary file 3. List of 657 genes significantly upregulated in NvPOU4 mutants relative to their siblings.

• Supplementary file 4. List of Gene Ontology terms overrepresented in genes downregulated in NvPOU4 mutants relative to their siblings.

- Supplementary file 5. List of Gene Ontology terms overrepresented in genes upregulated in NvPOU4 mutants relative to their siblings.
- Supplementary file 6. List of 293 POU-IV-activated genes.
- Supplementary file 7. List of 178 POU-IV-repressed genes.
- Supplementary file 8. List of Gene Ontology terms overrepresented in POU-IV-activated genes.
- Supplementary file 9. List of POU-IV downstream target genes represented in the adult metacell c79 (hair cell).
- Supplementary file 10. List of Gene Ontology terms overrepresented in POU-IV-activated genes represented in the adult metacell c79 (hair cell).
- Supplementary file 11. List of POU-IV downstream target genes represented in the cnidocyte metacell c8.
- Supplementary file 12. List of Gene Ontology terms overrepresented in POU-IV-repressed genes represented in the cnidocyte metacell c8.
- Supplementary file 13. Lists of POU-IV downstream target genes represented in POU-IV-positive adult metacells c63, c64, c65, c66, c75, c76, c100, c101, and c102.
- Supplementary file 14. List of GPCR-encoding genes in the adult metacell c79 (hair cell).
- Supplementary file 15. List of gene-specific primer sequences used to amplify *polycystin 1* cDNA.
- Transparent reporting form

## Data availability

Sequencing data have been deposited in GenBank under an accession number OK338071, and in BioProject database under an accession number PRJNA767103. The scripts for RNA-Seq and ChIP-seq analysis are publicly available at https://github.com/pyrosilesl97/POU-IV_analysis (copy archived at swh:1:rev:463178242d112edab7094c12e093dd780177885b).

The following datasets were generated:

| Author(s) | Year | Dataset title | Dataset URL | Database and Identifier |
|---|---|---|---|---|
| Nakanishi N | 2021 | Cnidarian hair cell development illuminates an ancient role for the class IV POU transcription factor in defining mechanoreceptor identity | https://www.ncbi.nlm.nih.gov/bioproject/?term=PRJNA767103 | NCBI BioProject, PRJNA767103 |
| Nakanishi N | 2021 | *Nematostella vectensis* polycystin-1 mRNA, complete cds | https://www.ncbi.nlm.nih.gov/nuccore/OK338071 | NCBI Nucleotide, OK338071 |

The following previously published dataset was used:

| Author(s) | Year | Dataset title | Dataset URL | Database and Identifier |
|---|---|---|---|---|
| Rentzsch F | 2020 | RNA-seq of Nematostella POU4 mutants at primary polyp stage | https://www.ebi.ac.uk/arrayexpress/experiments/E-MTAB-8658/ | ArrayExpress, E-MTAB-8658 |

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
