## [Editor Report]

This study focusses on a little understood cell type, the hair cell , in the sea anemone *Nematostella vectensis*. It shows that the POU-IV transcription factor is required for the maturation of these likely mechanosensory neurons and activates a wide array of mechanosensory effector proteins. Because POU-IV transcription factors also play essential roles for the differentiation of mechanoreceptors in many bilaterian phyla, this suggests an evolutionarily ancient role of POU-IV in regulating mechanosensory identity.

---

## [Decision Letter]

**Decision letter after peer review:**

Thank you for submitting your article "Cnidarian hair cell development illuminates an ancient role for the class IV POU transcription factor in defining mechanoreceptor identity" for consideration by *eLife*. Your article has been reviewed by 3 peer reviewers, and the evaluation has been overseen by a Reviewing Editor and Claude Desplan as the Senior Editor. The following individuals involved in review of your submission have agreed to reveal their identity: Bernd Fritzsch (Reviewer #1); Gerhard Schlosser (Reviewer #2); Jr-Kai Yu (Reviewer #3).

Essential revisions:

1) There were some possible concerns about the specificity of the POU4 antibody staining. Adding a pou4 RNA in situ either with co-staining of the POU4 antibody or a side-by-side staining will make the antibody staining significantly more trustworthy. Given that all technical aspects of this experiment are well-established, we believe that it should not be too demanding.

2) The addition of phylogenetic trees of bHLH factors that in other organisms function upstream in the regulatory cascade of pou4 (specifically the atonal/ath family) and a speculation about their possible function in the context of the nematostella hair cell type will provide interesting conceptual value to the manuscript.

3) Along the same lines, the question on the sensory modalities of these 'hair cells', including possible multimodularity, should be included in the discussion.

*Reviewer #2 (Recommendations for the authors):*

The study is overall very well presented and carefully discussed. I have only two comments.

First, in line 999 and following, the authors infer that the presence of a relatively large number of repressed genes in cnidocytes suggests that POU acts mainly as a repressor in these cells required to fine tune the expression levels of these target genes. But the same finding is open to alternative interpretations. The fact that is a large number of POU-IV repressed genes is expressed in cnidocytes could also mean that POU4 is not sufficient to repress these genes specifically in cnidocytes because it requires a co-repressor which is specifically absent in cnidocytes.

Second, based on morphology and some behavioral data, the authors assume throughout the manuscript that the hair cells are dedicated mechanoreceptors. However, it is equally possible that these cells are multimodal and have additional chemosensory (and/or even photosensory) functions since multimodal cells seem to be common in cnidarians. So a more careful discussion is required here. The authors should explore the transcriptome data more; maybe they provide some insight into the sensory modalities mediated by these cells.

*Reviewer #3 (Recommendations for the authors):*

1. About the specificity of the POU-IV antibody ("Weakness" #1), I would suggest the authors to use the antigen sequence to search against the *N. vectensis* genome, to check whether there are any significant matches to other annotated proteins than the POU-IV factor. In addition, double staining (in situ hybridization plus immunostaining) should be helpful to verify their results.

2. As mentioned in my public review section ("Weakness" #2), the authors observed clear reduction of the phalloidin staining signals underneath the surface of the putative hair cells in pou-iv -/- polyps, however, from their EM pictures, it is not clear to me whether the rootlet structure of the apical cilium might be also affected by the loss of POU-IV function. Specifically, in their Figure 5K and 5L, the area encircled by the dotted purple line (which is really difficult to recognize!) seem to contain the structure of cilium rootlet, but it looks less prominent compared to that shown in Figure 1l. I hope the authors can provide better evidence to support their assessment on the rootlet structure phenotype.

---

## [Author Response]

Essential revisions:1) There were some possible concerns about the specificity of the POU4 antibody staining. Adding a pou4 RNA in situ either with co-staining of the POU4 antibody or a side-by-side staining will make the antibody staining significantly more trustworthy. Given that all technical aspects of this experiment are well-established, we believe that it should not be too demanding.

As suggested, we have added a supplemental figure (Figure 2—figure supplement 1) which shows that the pattern of immunostaining with the anti-POU-IV antibody recapitulates that of *pou-iv* mRNA in situ hybridization signals.

2) The addition of phylogenetic trees of bHLH factors that in other organisms function upstream in the regulatory cascade of pou4 (specifically the atonal/ath family) and a speculation about their possible function in the context of the nematostella hair cell type will provide interesting conceptual value to the manuscript.

It is indeed possible that bHLH factors function upstream of POU-IV in the context of hair cell development – we have included a discussion about this possibility in the revised manuscript. As phylogenetic analyses of cnidarian bHLH factors have been previously carried out (e.g. Simionato et al., 2007), we chose to cite the relevant work in the revision instead of carrying out the analyses ourselves. These prior studies indicate that cnidarians lack unambiguous *atonal* orthologs*,* but have several divergent bHLH genes of the Atonal superfamily that consists of multiple bHLH gene families including Atonal, neurogenin and NeuroD. We have stressed the relevance of examining the potential roles of these atonal-like bHLH genes in cnidarian hair cell development to understanding the evolution of gene regulatory mechanisms underpinning mechanoreceptor development across Cnidaria and Bilateria.

3) Along the same lines, the question on the sensory modalities of these 'hair cells', including possible multimodularity, should be included in the discussion.

We agree that hair cells could be multimodal sensory cells, and have included a discussion that raises this possibility in the revised version of the manuscript.

Reviewer #2 (Recommendations for the authors):The study is overall very well presented and carefully discussed. I have only two comments.First, in line 999 and following, the authors infer that the presence of a relatively large number of repressed genes in cnidocytes suggests that POU acts mainly as a repressor in these cells required to fine tune the expression levels of these target genes. But the same finding is open to alternative interpretations. The fact that is a large number of POU-IV repressed genes is expressed in cnidocytes could also mean that POU4 is not sufficient to repress these genes specifically in cnidocytes because it requires a co-repressor which is specifically absent in cnidocytes.

Indeed, it would be more accurate to view POU-IV as a leaky repressor in cnidocytes, and this could very well be because of the absence of a co-repressor in this cell type. We have slightly modified the language to clarify this point.

Second, based on morphology and some behavioral data, the authors assume throughout the manuscript that the hair cells are dedicated mechanoreceptors. However, it is equally possible that these cells are multimodal and have additional chemosensory (and/or even photosensory) functions since multimodal cells seem to be common in cnidarians. So a more careful discussion is required here. The authors should explore the transcriptome data more; maybe they provide some insight into the sensory modalities mediated by these cells.

We agree that hair cells could be multimodal sensory cells, and have included a discussion that raises this possibility in the revised manuscript. As suggested, we have examined the metacell data, and added a list of GPCRs – possible chemosensory receptors – represented in the hair-cell metacell as a supplementary file.

Reviewer #3 (Recommendations for the authors):1. About the specificity of the POU-IV antibody ("Weakness" #1), I would suggest the authors to use the antigen sequence to search against the *N. vectensis* genome, to check whether there are any significant matches to other annotated proteins than the POU-IV factor. In addition, double staining (in situ hybridization plus immunostaining) should be helpful to verify their results.

We thank the reviewer for these suggestions. Accordingly, we have performed TBLASTN search of the antigen sequence against the Nematotella genome, which yielded a single hit at the *pou-iv* locus. We have included this result in the Materials and methods section. Also, we have added *Pou-iv* in situ data (Figure 2—figure supplement 1) that confirm comparability between the pattern of *pou-iv* mRNA expression and that of POU-IV antibody staining.

2. As mentioned in my public review section ("Weakness" #2), the authors observed clear reduction of the phalloidin staining signals underneath the surface of the putative hair cells in pou-iv -/- polyps, however, from their EM pictures, it is not clear to me whether the rootlet structure of the apical cilium might be also affected by the loss of POU-IV function. Specifically, in their Figure 5K and 5L, the area encircled by the dotted purple line (which is really difficult to recognize!) seem to contain the structure of cilium rootlet, but it looks less prominent compared to that shown in Figure 1l. I hope the authors can provide better evidence to support their assessment on the rootlet structure phenotype.

We have added a supplementary figure showing a ciliary rootlet in the putative hair cell of a *pou-iv* mutant. Also, we have made the dotted purple lines in Figure 5K, L thicker.